# AutoMat: Physics-Guided Agentic Reasoning for Solving Ill-Posed Inverse Microscopy Problems

Yaotian Yang [* 1]   Yiwen Tang [* 2]   Yizhe Chen [* 1]   Xiao Chen [1 3]   Jiangjie Qiu [1]   Hao Xiong [1]   Haoyu Yin [1]
Zhiyao Luo [1]   Yifei Zhang [1]   Sijia Tao [1]   Wentao Li [1]   Qinghua Zhang [1]   Yuqiang Li [2]   Wanli Ouyang [2]   Bin Zhao [2]
Xiaonan Wang [1]   Fei Wei [1 3]

## Abstract

Reconstructing atomistic crystal structures from a single noisy STEM projection is an ill-posed inverse problem: multiple lattices can explain similar contrast, and purely feed-forward models cannot verify physical validity. We present **AutoMat**, a failure-aware agentic *controller* that performs inference-time hypothesis search with *closed-loop verification* to convert Scanning Transmission Electron Microscopy (STEM) images into simulation-ready crystal structures and downstream properties. AutoMat composes perception and physics modules—pattern-adaptive denoising, physics-guided template retrieval *as a state-dependent auxiliary branch*, symmetry-constrained atomic reconstruction, and MLIP-based relaxation/validation—and triggers rollback-and-retry when verification fails. For systematic evaluation, we introduce **STEM2Mat-Bench**, a benchmark dataset containing 450+ annotated samples. Performance is assessed using lattice root-mean-square deviation (RMSD), formation energy mean absolute error (MAE), and structure matching accuracy. Results demonstrate that AutoMat outperforms existing approaches including SOTA models, specialized domain tools, and closed-source multimodal large models. This work establishes a direct pathway from microscopic characterization to atomic-scale modeling, addressing a fundamental challenge in materials science.

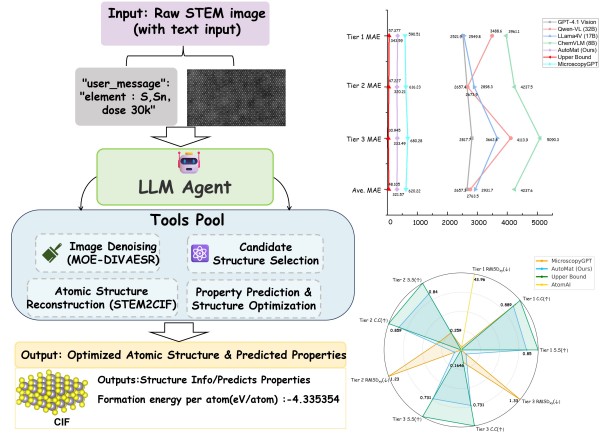

**Figure 1. Overview of AutoMat.** The left part shows a failure-aware reasoning *controller* that composes four modules—pattern-adaptive denoising, template selection (state-triggered auxiliary branch), atomic reconstruction (STEM2CIF), and MLIP-based relaxation/property prediction—into a closed-loop verification workflow. The right panel presents a line chart and radar plot comparing different models in terms of energy and structural errors.

*Equal contribution [1]Department of Chemical Engineering, Tsinghua University, Beijing, China [2]Shanghai Artificial Intelligence Laboratory, Shanghai, China [3]Ordos Laboratory, Ordos, China. Correspondence to: Xiao Chen <chenx123@tsinghua.edu.cn>, Bin Zhao <bin@nwpu.edu.cn>, Xiaonan Wang <wangxiaonan@tsinghua.edu.cn>, Fei Wei <wf-dce@tsinghua.edu.cn>.

*Proceedings of the $43^{rd}$ International Conference on Machine Learning*, Seoul, South Korea. PMLR 306, 2026. Copyright 2026 by the author(s).

## 1. Introduction

Machine learning interatomic potentials and force fields now approach *ab initio* accuracy in predicting atomic energies and forces (Yang et al., 2024; Batatia et al., 2024; Gasteiger et al., 2021; Liao et al., 2024; Lu et al., 2023) but remain limited by experimental validated crystal structures. Meanwhile, Scanning Transmission Electron Microscopy (STEM) can image atoms at sub-ångström resolution (Zhang et al., 2018; 2023; 2020; Ortalan et al., 2010), yet translating into quantitative crystal structures still relies on expert-driven, time-consuming annotation. This creates a gap between structural representation and theoretical validation in materials science (Kalinin et al., 2022; 2023).

Although recent advances in STEM image analysis have shown promise, most existing studies focus on individual components like denoising (Yang et al., 2025; Lin et al., 2020; 2021; Wang et al., 2020), atom localization (Ziatdinov et al., 2017; Borshon et al., 2024; Eliasson & Erni, 2024), reconstruction (De Backer et al., 2022; Stoops et al., 2025;

Huang et al., 2025) and phase classification (Li et al., 2021; Maksov et al., 2019). These approaches remain fragmented and are not integrated into end-to-end system. Conventional image denoising techniques suppress noise and improve contrast at the pixel level but cannot yield periodic or chemically meaningful crystal structures (Gambini et al., 2023; Ihara et al., 2022; Wang et al., 2020; Yang et al., 2025). Atomic detection models can localize atomic peaks but cannot infer complete lattices[1] or identify atomic species. General-purpose multimodal models like GPT-4.1mini (Achiam et al., 2023) and Qwen2.5-VL (Bai et al., 2025) exhibit basic image understanding but lack the ability to produce simulation-ready formats such as crystallographic file (CIF). Even domain-specific tools like AtomAI (Ziatdinov et al., 2022) and the newly released domain SOTA model MicroscopyGPT (Choudhary, 2025) can only predict atomic coordinates and structural descriptions for minimal systems, without supporting complete CIF structure reconstruction or property prediction. Meanwhile, public datasets (e.g., JARVIS-STM) mainly target STM (Scanning Tunneling Microscopy) images, lack DFT-level energy labels, and are unsuitable for benchmarking structure–property pipelines. As a result, the field lacks a fully automated end-to-end system that can convert raw STEM images into reconstructed structures and simulated properties, along with the standardized benchmark for comprehensive evaluation.

To address this gap, as shown in Fig. 1, we introduce **AutoMat**, a failure-aware agentic *controller* that links STEM imaging with atomistic simulation via inference-time hypothesis search and closed-loop verification, along with a benchmark tailored to the task. At the core of AutoMat is a controller equipped with a suite of four modular tools, which it can dynamically compose to solve the task: 1) **Pattern-Adaptive STEM Image denoising**: We apply a pattern-adaptive mixture-of-experts network, MOE-Denoising Inference Variational Autoencoder Super-Resolution (Yang et al., 2025)(MOE-DIVAESR), to denoise and enhance raw iDPC-STEM[2] images. The ResNet-18-based gating network elects the most suitable expert network for each input image based on its estimated noise level, enabling joint denoising, inpainting, and super-resolution. 2) **Physics-Guided Template Retrieval**: Enhanced images are matched to a large-scale library of simulated STEM projections. Top candidate structures are retrieved using pixel similarity and filtered by elemental contrast patterns to produce strong structural priors. 3) **Symmetry-Constrained Structure Reconstruction**: Atomic peaks are detected via unsupervised clustering. We fit the lattice under symmetry constraints,

assign atomic species based on the candidate, and generate the standard CIF file representing the periodic crystal structure. 4) **Energy Evaluation via Machine-Learned Potential**: The reconstructed structure is relaxed using the pretrained MatterSim potential to predict formation energy. The language agent autonomously orchestrates the workflow, planning tool execution and adaptively retrying failed steps based on quality checks.

To support rigorous evaluation, we curated 2,143 high-quality monolayer structures from C2DB (Haastrup et al., 2018), Materials Project (Jain et al., 2013), and OpenCrystal (Vaitkus et al., 2023), and simulated their corresponding iDPC-STEM images with abTEM (Madsen & Susi, 2020). From this pool we selected 459 representative image–structure pairs, which constitute our **STEM2Mat** benchmark and are used for all subsequent evaluations. Our evaluation metrics include projected lattice RMSD, formation energy MAE, and atom-wise structure matching success rate. Additionally, we introduce three fine-grained indicators to assess reconstruction quality (e.g., atomic recall/precision), robustness across noise levels, and computational efficiency. On this benchmark, AutoMat achieves a projected RMSD of $0.11 \pm 0.03$ Å, energy MAE below 350 meV/atom, and an atomic correspondence total success rate of 83.2%, outperforming GPT-4.1-mini, Qwen-VL, LLama4V, ChemVLM (Li et al., 2025), MicroscopyGPT and AtomAI by an order of magnitude. These results demonstrate AutoMat as a reproducible and accurate end-to-end solution for microscopy-driven materials modeling.

The contributions of this paper are summarized as follows:

• **Physics-Aware Reasoning Controller**. We formulate STEM-to-structure reconstruction as inference-time hypothesis search with *closed-loop verification*, and introduce AutoMat as a controller that composes denoising, a state-dependent retrieval-guided prior branch, reconstruction, and relaxation with failure signals and rollback-and-retry.

• **Algorithmic advancements**. We design MOE-DIVAESR as a pattern-adaptive denoiser for diverse STEM images, enabling efficient noise reduction, defect correction, and detail enhancement. We also develop STEM2CIF to reconstruct crystal structures by identifying the minimal repeating unit using symmetry heuristics and physical constraints, then converting the result into standard CIF format.

• **STEM2Mat benchmark & evaluation suite**. We curate 2,143 distinct monolayer structures to simulate large-field iDPC-STEM images, and provide a benchmark split with unambiguous evaluation protocols and unified metrics for lattice reconstruction accuracy, energy fidelity, robustness, and computational efficiency.

• **Reproducible end-to-end results**. We show that AutoMat substantially improves both structural fidelity and

---

[1]In crystallography, a lattice refers to the periodic arrangement of atoms in a crystal structure.

[2]iDPC-STEM stands for integrated Differential Phase Contrast Scanning Transmission Electron Microscopy, a technique that enhances light element contrast in atomic-resolution imaging.

downstream energy prediction compared with off-the-shelf multimodal baselines. Code, data, and evaluation tools are publicly available at `https://github.com/yyt-2378/AutoMat` and `https://huggingface.co/datasets/yaotianvector/STEM2Mat` to support reproducibility.

## 2. Related Work

**Automated Microscopy Image Analysis.** In recent years, deep learning methods have been extensively applied to electron microscopy and scanning probe microscopy data analysis (Yang et al., 2025; Lin et al., 2020; 2021; Wang et al., 2020; Ziatdinov et al., 2017; Borshon et al., 2024; De Backer et al., 2022; Stoops et al., 2025; Huang et al., 2025). Current approaches range from unsupervised defect detection to supervised atomic column identification. AtomAI (Ziatdinov et al., 2022), for example, integrates microscopy images with computational simulations, but primarily focuses on atom segmentation and identification. More recently, SciLink (Yao et al., 2025) proposes an open-source *multi-agent* framework that closes the loop between microscopy/spectroscopy experiments, literature-based novelty assessment, and theory-in-the-loop simulations, and has demonstrated impressive performance in automated defect localization and atomic-scale analysis. However, its primary focus is on serendipity-aware, high-level experiment–theory orchestration. It does not yet provide a dedicated solution for reconstructing electron-microscopy images into explicit crystal structures that can be quantitatively compared with theoretical models, nor for using such reconstructed structures as direct inputs to downstream simulations and property prediction.

**STEM Image to Structure Reconstruction.** Existing methods for reconstructing crystal structures from STEM images typically rely on multiple images, prior structural information, or are limited to single-element systems. For instance, De Backer et al. (De Backer et al., 2022) employed Bayesian algorithms for 3D reconstruction, but their approach was demonstrated only for simple single-element systems. Currently, few methods can generate standard crystallographic information files (CIF) from single experimental images, especially for complex multi-element 2D crystals (Stoops et al., 2025).

**Vision-Language Models in Chemistry.** Recently, multimodal large language models (ChemVLM (Li et al., 2025), GPT-4.1mini (Achiam et al., 2023), and Qwen-VL (Bai et al., 2025)) have started to be explored in chemistry and materials science. However, these models generally lack the capability to accurately handle detailed spatial structure tasks, and recent benchmarks further reveal that vision-language models can exhibit erroneous reasoning in visually grounded tasks (Shi et al., 2026). Existing chemical agent tools (e.g., ChemCrow (Bran et al., 2023) and Chemagents (Tang et al., 2025a)) are limited to text-based descriptions and cannot process image-based inputs, significantly restricting their applicability in microscopy-based analyses. The field's SOTA multimodal model Microscopy-GPT (Choudhary, 2025) can generate structural descriptions from STEM images but cannot yet reconstruct CIFs or predict properties.

**Material Property Prediction Models.** Advances in machine learning-based interatomic potentials (such as Matter-Sim (Yang et al., 2024), M3GNet (Chen & Ong, 2022), and MACE (Batatia et al., 2023)) have significantly improved the accuracy of computational property predictions. Combining these models with experimental imaging provides a novel, digital twin-like approach for validating structural reconstructions. However, existing benchmarks predominantly evaluate models using theoretical datasets, lacking end-to-end assessments starting from experimental image inputs.

In summary, these approaches highlight progress and limitations in microscopy image analysis, structure reconstruction, multimodal modeling, and property prediction. **AutoMat addresses these gaps** by integrating an agent from STEM images to material property prediction and establishes the STEM2Mat benchmark for evaluating the robustness and scalability of automated material characterization.

## 3. Dataset Construction

### 3.1. Composition and Taxonomy

Our benchmark focuses on two-dimensional (2D) materials, whose atomic-scale thickness allows STEM to resolve individual columns with minimal multiple-scattering artifacts. Starting from nearly 10,000 candidate structures harvested from C2DB, Materials Project, and OpenCrystal,[3] we followed a two-stage curation process. First, automated filters removed non-stoichiometric, partially occupied, or 3D bulk entries. Second, domain experts inspected symmetry, cleavage energy, and substrate feasibility, yielding **2,143** high-confidence monolayer crystals. The collection spans six chemical families (Fig. 5): *(i)* classic 2D materials—graphene, $MoS_2$, h-BN, black phosphorus; *(ii)* emergent allotropes, e.g., silicene, borophene; *(iii)* conductive MX-enes (23 distinct formulas); *(iv)* intrinsic 2D magnets such as $CrI_3$ and $Fe_3GeTe_2$; *(v)* Janus structures typified by MoSSe; and *(vi)* Ruddlesden–Popper–type 2D perovskites. Elemental diversity is broad: 67 unique elements appear, producing 76 unary, 1,409 binary, and 658 ternary systems. Each structure is stored as a CIF file with validated lattice vectors and fractional atomic coordinates.

---

[3]All sources were queried in April 2025 using identical monolayer filters.

## 3.2. Image Simulation and Data Augmentation

To simulate realistic large-field STEM imaging conditions, we generated synthetic iDPC-STEM micrographs using the open-source `abTEM` simulation engine. For each structure, a random $12 \times 12$ to $16 \times 16$ supercell was constructed and projected at $0.1\,\text{Å/pixel}$ resolution. Five electron-dose settings ($1$–$6 \times 10^4\,e^-/\text{Å}^2$) and realistic lens aberrations were sampled to mimic experimental conditions. Poisson detector noise was injected to match reported signal-to-noise ratios. To study model robustness, we applied Gaussian blurring and dose-specific shot noise to simulate additional imaging imperfections. Ground-truth atomic coordinates were rendered into Gaussian "atom masks" to enable supervised training of localisation models. Each sample thus forms an image–structure–property triplet: (i) the noisy STEM projection, (ii) the corresponding ground-truth CIF, and (iii) DFT-level formation energy (along with band gap and magnetic moment, if available). We conducted principal component analysis (PCA) on structural fingerprints, which revealed clear clustering patterns. Subsequently, $k$-means clustering ($k=2, 6$) was used to ensure balanced train/validation/test splits across the chemical diversity of the dataset.

## 3.3. STEM2Mat Benchmark Split and Tiering

To construct a representative and tractable benchmark dataset for STEM-based crystal modeling, we applied stringent geometric and chemical screening criteria to the 2,143 collected 2D material structures, ensuring their suitability for monolayer imaging and end-to-end reconstruction. Specifically, we retained only structures containing no more than three distinct elements. For those with multiple elements, we required a minimum atomic-number span of ten, i.e., $\max(Z_i) - \min(Z_i) \geq 10$, to ensure sufficient imaging contrast between heavy and light atoms. To guarantee monolayer geometry, we limited the $z$-axis thickness to no more than 3 Å. Each structure's atomic coordinates were then projected onto the $(x, y)$ plane, discretized onto a 1 Å grid, and evaluated for overlapping projections. Only structures with a projected duplication ratio below 10%, i.e., $\frac{\text{Number of overlapping grid points}}{\text{Total grid points}} \leq 0.1$, were retained to avoid ambiguity in atomic interpretation.

Following this multi-criteria filtering process, we retained exactly **459 unique** well-defined, unambiguous monolayer structures—21% of the original dataset—for blind end-to-end evaluation. The remaining 1,693 samples were split into training (80%) and validation (20%) sets to support model training and tuning.

To analyze model performance as a function of task difficulty, we generated a total of **570 test instances** from these 459 unique structures. The tiers are stratified based on material composition and imaging noise (see Fig. 2):

• **Tier 1** (35 instances) – A subset of unary materials simulated under high electron doses ($5$–$6 \times 10^4\,e^-/\text{Å}^2$); these images are high contrast, low noise, and represent the easiest cases.

• **Tier 2** (459 instances) – **The complete set of 459 unique structures** simulated under moderate electron dose conditions ($3$–$4 \times 10^4\,e^-/\text{Å}^2$); this tier covers the full diversity of the test set with intermediate complexity.

• **Tier 3** (79 instances) – A subset of ternary compounds or complex structures imaged at low dose ($1$–$2 \times 10^4\,e^-/\text{Å}^2$); these samples exhibit high noise, complex contrast, and are the most challenging to reconstruct.

This design allows us to evaluate robustness against signal degradation on identical underlying topologies. With the STEM2Mat-Bench tiers defined, we now specify the quantitative metrics used throughout the paper. For detailed definitions and formulae of the evaluation metrics, see Section 3.4.

## 3.4. Evaluation Metrics

To compare methods reproducibly, we report two *primary* metrics—energy error and lattice error—and two *holistic* metrics that reflect chemical and structural validity.

**Mean Absolute Error (MAE).** Average difference between predicted and DFT formation energies. We note that Formation Energy MAE comprises two error sources: (i) geometric reconstruction error, and (ii) intrinsic MLIP prediction error. Since the MLIP error is constant across all baselines (and bounded by the Oracle Upper Bound), the significant MAE reduction achieved by AutoMat is attributable to superior structural reconstruction accuracy.

$$\text{MAE} = \frac{1}{N} \sum_{i=1}^{N} \left| E_i^{\text{pred}} - E_i^{\text{ref}} \right| \; [\text{meV/atom}] \qquad (1)$$

**Projected Lattice RMSD.** Deviation of in-plane lattice constants:

$$\text{RMSD}_{xy} = \sqrt{\frac{1}{2}\left[ (a^{\text{pred}} - a^{\text{ref}})^2 + (b^{\text{pred}} - b^{\text{ref}})^2 \right]} \qquad (2)$$

**Composition Correctness (C.C.).** 1 if elemental types and counts match the reference; 0 otherwise.

**Structure Success Rate (S.S.).** A prediction is successful when it satisfies both chemistry and geometry. First define 2-D spatial similarity

$$S_{\text{spatial}} = \exp\left(-\text{MSE}_{\text{2D}}\right), \qquad (3)$$

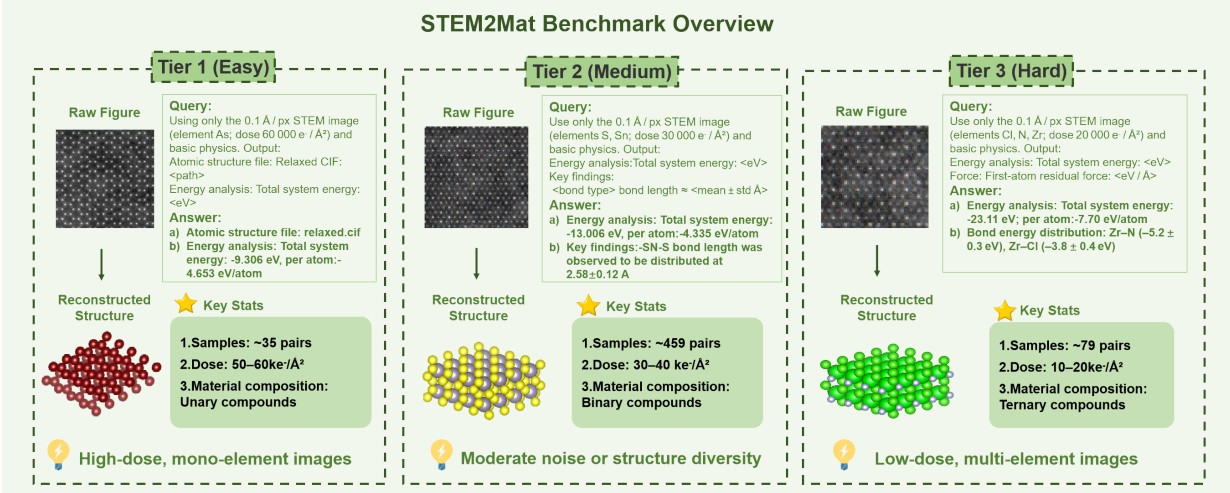

*Figure 2.* **Overview of the STEM2Mat benchmark design**, illustrating the tiered classification of STEM samples by material complexity and imaging dose, which systematically stratifies reconstruction difficulty from simple unary to complex ternary compounds.

where $\mathrm{MSE}_{2D}$ is the mean-squared error of projected atomic positions after optimal element-wise matching. Then

$$\mathrm{S.S.} = \frac{1}{N} \sum_{i=1}^{N} \mathbb{I}\Big[ S_{\mathrm{spatial}}^{(i)} \geq 0.8 \Big] \times 100\%. \quad (4)$$

## 4. Overview of AutoMat

### 4.1. Limitations of Existing Large Multimodal Models and Domain Tools

Most current large multimodal models (e.g., GPT-4.1mini, LLama4V, Qwen2.5-VL, ChemVLM) focus on general-purpose image understanding tasks such as scene recognition, OCR, and molecular structure identification, but exhibit limited capability in interpreting scientific images like electron microscopy (EM). Domain-specific chemistry agents (e.g., ChemAgent, ChemCrow) remain predominantly text-based, executing tool calls for predefined tasks without closed-loop visual reasoning or image-guided decision-making. Specialized STEM data toolkits (e.g., AtomAI) provide atomic coordinate extraction and segmentation, yet their applicability is largely restricted to single-element nanoparticles and fail with complex, multi-element STEM images. MicroscopyGPT, the current state-of-the-art multimodal model in the domain, can generate structural descriptions from user-provided STEM images but still relies on known structural coordination and cannot directly reconstruct CIFs.

To date, existing agents are unable to autonomously generate simulation-ready CIFs and predict formation energies from a single real STEM image, thus completing a generalizable structure–property pipeline. To bridge this gap, we introduce **AutoMat**, an agent framework with advanced reasoning, enabling end-to-end modeling from pixel-level inputs to material property predictions.

### 4.2. Tools Pool

Within AutoMat—a general and efficient framework—we employ four core tools: MOE-DIVAESR, Image Template Matching, STEM2CIF, and MatterSim, responsible for image denoising, template matching, structure reconstruction, and property prediction, respectively. Further details are provided in Appendices A.2 and A.3.

**MOE-DIVAESR** is a structure-pattern–adaptive mixture-of-experts (MoE) model for STEM image denoising. Trained on an augmented STEM dataset, it contains multiple expert networks, each specializing in a distinct structural pattern, while a gating network dynamically selects the appropriate experts based on the input image features. This design enables denoising, defect correction, and fine-detail enhancement, producing sharply resolved atomic columns.

**Image Template Matching** compares an enhanced image against a database of known structures. By combining image features with elemental information, this technique efficiently narrows down potential candidates and identifies the structure that best fits experimental data. The process is even faster and more reliable when researchers can directly match against pre-calculated or known candidate structures.

**STEM2CIF** reconstructs a high-fidelity atomic model from the enhanced image, optionally conditioned on a matched template when the retrieval branch is invoked. It locates atomic column positions, infers lattice parameters, and identifies atomic species. Crystallographic symmetry heuristics and physical constraints then reduce the model to its minimal repeating unit, yielding a crystallographic CIF file.

**MatterSim** is a pretrained machine-learning interatomic potential for rapid structural relaxation and property evaluation. Trained on large-scale DFT datasets, it attains near-DFT accuracy for energies and properties. Integrated with

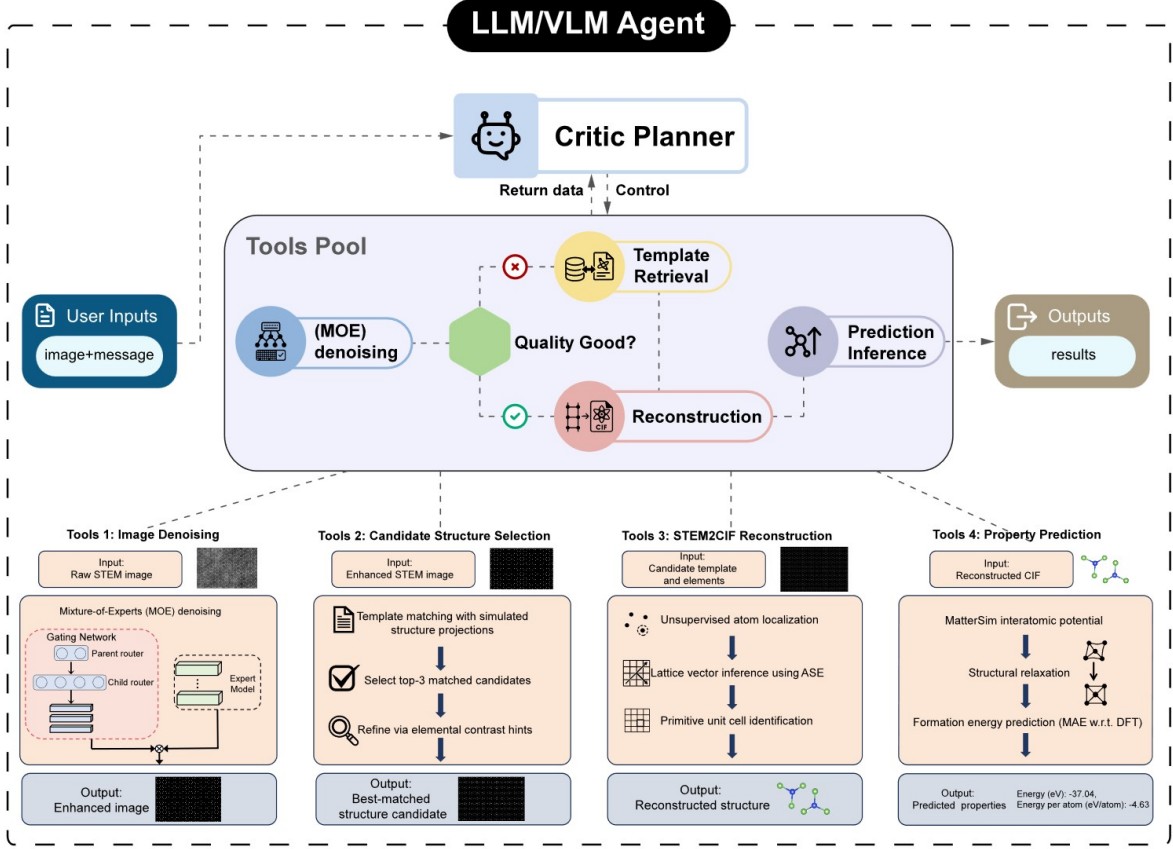

*Figure 3.* **AutoMat's reasoning controller composes four stages**—denoising, template matching, structure reconstruction, and MLIP-based relaxation/property prediction—from STEM image to relaxed crystal with properties via closed-loop verification.

ASE, it enables structure optimization and fast estimation of quantities such as total energy, formation energy, and elastic moduli, providing a swift and accurate alternative to conventional DFT.

### 4.3. Flexible Tool-Calling Framework

**Formulation.** We formulate reconstruction as a search process over a state space. **State** $s$: current intermediate structure (e.g., denoised image, retrieved template, candidate CIF); **Action** $a$: a tool execution (e.g., `Denoise`, `Retrieve`, `Refine`); **Verification** $\mathcal{V}(s)$: a physics-based critic (e.g., consistency checks via relaxation/simulation) that emits a failure signal when the current hypothesis is physically invalid; **Policy** $\pi(a \mid s)$: a controller that triggers **rollback-and-retry** upon failure, effectively pruning invalid branches and performing inference-time search.

**Physics-Guided Reasoning Loop.** Building on this formulation, **AutoMat** establishes a decision control loop (Fig. 3) where the LLM backbone (e.g., **DeepSeekV3**) acts as a *text-based reasoning controller*. Instead of processing pixels directly, it reasons over structured tool outputs **(e.g., quality metrics)** to re-plan the workflow dynamically. In the execution phase, the policy $\pi$ adapts to the input state: First, **MOE-DIVAESR** is called for pattern-adaptive denoising; if quality thresholds are not met, this step is repeated. Next, the agent invokes **STEM2CIF** to localize atoms, fit lattices, and output confidence scores. **Image Template Matching** is not a mandatory first step; if the denoising quality is sufficient, the controller will skip retrieval. Retrieval serves as a *state-dependent auxiliary branch*, triggered only when the direct path fails or a stronger structural prior is needed. Finally, **MatterSim** relaxes the structure and predicts properties for **physical validation**. By monitoring intermediate results and executing rollbacks upon failure, AutoMat achieves robust end-to-end automation without relying on any single model.

## 5. Experiments and Results

### 5.1. Quantitative Evaluation

**Baseline Overview.** To make comparisons fair and interpretable, we group baselines by *task scope*:

- **Image → Property (no explicit structure).** GPT-4.1mini, Qwen2.5-VL (32B), LLama4V (17B),

*Table 1.* **Formation-energy MAE (meV/atom) across tiers.** Lower is better ($\downarrow$). "–" indicates the method does not provide an energy prediction.

| Method | Tier 1 $\downarrow$ | Tier 2 $\downarrow$ | Tier 3 $\downarrow$ | Avg. $\downarrow$ |
|---|---|---|---|---|
| GPT-4.1 Vision | 2521.9 | 2657.4 | 2817.7 | 2657.3 |
| Qwen-VL (32B) | 3488.6 | 2673.9 | 4113.9 | 2763.5 |
| LLama4V (17B) | 2549.8 | 2898.3 | 3662.6 | 2931.7 |
| ChemVLM (8B) | 3961.1 | 4237.5 | 5090.3 | 4237.6 |
| MicroscopyGPT (11B) | 590.51 | 616.23 | 680.28 | 620.22 |
| AtomAI | – | – | – | – |
| **AutoMat (Ours, DeepSeek)** | **343.59** | **320.21** | **333.49** | **321.57** |
| **AutoMat (Ours, GPT-4o)** | **341.72** | **322.05** | **334.12** | **323.25** |
| Upper Bound | 57.377 | 47.227 | 30.945 | 48.105 |

ChemVLM (8B) and MicroscopyGPT (11B) receive a fixed prompt, composition hints, and a STEM image to infer material properties. These baselines do *not* produce simulation-ready CIFs, so we only compare them on energy metrics.

- **Image → Structure (structure-only tools).** AtomAI's segmentation network detects atomic centers; relative coordinates plus image resolution are used to fit the lattice. This baseline outputs atomic models and is evaluated on structural metrics.

- **Structure → Property (oracle bound).** The ground-truth CIF is fed directly to the MatterSim MLIP to benchmark formation-energy error under perfect-structure assumptions (MLIP-bias bound).

We summarize the performance of these baselines and our **AutoMat** system on the test set in Tables 1 and 2, which together cover samples across Tier 1–3 difficulty levels. To further separate tool-level capability from controller-level orchestration, we also evaluate same-tool scripted variants. In the main text, we report a stronger heuristic script with hand-coded quality gating, retrieval branching, rollback, and bounded retries. Detailed ablations and backbone sensitivity analyses are provided in Appendices A.9 and A.4. Here we present results using DeepSeek (LLM) and GPT-4o (VLM) as representative backbone models; experiments show that AutoMat achieves comparable and consistent performance across different backbones.

**Discussion of Results.** For energy prediction, **AutoMat** achieves a mean formation energy MAE of $332 \pm 12$ meV/atom, with tier-wise results of 343.59, 320.21, and 333.49 meV/atom. Although this is higher than the MLIP lower bound of 57 meV/atom, it is still far better than the multi-eV errors of vision–language models. Even compared with MicroscopyGPT (used off-the-shelf under a fixed prompt and without additional fine-tuning on STEM2Mat-Bench), AutoMat remains much stronger, with Microscopy-GPT reaching only about 620 meV/atom. As task difficulty increases, most baselines show higher MAE, confirming the soundness of our tiered benchmark. These results indicate

*Table 2.* **Structural-accuracy metrics for methods that output atomic models.** $RMSD_{xy}\downarrow$: in-plane lattice RMSD (lower is better); C.C.$\uparrow$: composition correctness; S.S.$\uparrow$: structure success rate.

| Tier | Method | $RMSD_{xy}$ (Å)$\downarrow$ | C.C. (%)$\uparrow$ | S.S. (%)$\uparrow$ |
|---|---|---|---|---|
| 1 | AtomAI | 43.96±0.31 | 2.70 | 0.0 |
| | MicroscopyGPT | 1.56±0.72 | 92.8 | 0.0 |
| | **AutoMat (Ours, DeepSeek)** | **0.11±0.02** | **88.9** | **85.0** |
| | **AutoMat (Ours, GPT-4o)** | **0.11±0.02** | **89.2** | **85.3** |
| 2 | AtomAI | N/A | 0.0 | 0.0 |
| | MicroscopyGPT | 1.23±0.75 | 30.1 | 25.9 |
| | **AutoMat (Ours, DeepSeek)** | **0.11±0.03** | **85.9** | **84.0** |
| | **AutoMat (Ours, GPT-4o)** | **0.12±0.02** | **86.4** | **84.7** |
| 3 | AtomAI | N/A | 0.0 | 0.0 |
| | MicroscopyGPT | 1.33±0.92 | 16.46 | 0.0 |
| | **AutoMat (Ours, DeepSeek)** | **0.11±0.03** | **73.1** | **73.1** |
| | **AutoMat (Ours, GPT-4o)** | **0.11±0.02** | **72.8** | **72.8** |

*Table 3.* **Controller ablation against a stronger heuristic script.** The heuristic script uses the same tools as AutoMat with hand-coded quality gating, retrieval branching, rollback, and bounded retries. Energy MAE is reported in meV/atom. Lower is better for Energy MAE; higher is better for C.C. and S.S.

| Metric | Method | Tier 1 | Tier 2 | Tier 3 | Overall |
|---|---|---|---|---|---|
| Energy MAE $\downarrow$ | Heuristic Script | 380.00 | 373.00 | 585.00 | 385.42 |
| | **AutoMat** | **343.59** | **320.21** | **333.49** | **321.57** |
| C.C. (%) $\uparrow$ | Heuristic Script | 81.0 | 77.2 | 63.4 | 76.5 |
| | **AutoMat** | **88.9** | **85.9** | **73.1** | **83.2** |
| S.S. (%) $\uparrow$ | Heuristic Script | 79.4 | 74.8 | 62.3 | 74.2 |
| | **AutoMat** | **85.0** | **84.0** | **73.1** | **83.2** |

that AutoMat's remaining errors come mainly from reconstruction rather than MLIP limits, and that its predicted structures are reliable for downstream property evaluation. Figure 4 visualizes the reasoning gap: the top panel (Tier 3) shows that while baselines suffer from geometric hallucinations under low-dose conditions, AutoMat accurately recovers the lattice. The bottom panel demonstrates **zero-shot generalization** to real experimental ZSM-5 images, verifying robustness beyond the synthetic training distribution.

For structural reconstruction, AutoMat achieves an average projected $RMSD_{xy}$ of about 0.11 Å, much lower than MicroscopyGPT (1.3–1.6 Å, requiring prior coordination knowledge) and AtomAI (43–44 Å). Most deviations can be corrected through final relaxation. In composition correctness, AutoMat averages 83% across tiers (88.9%, 85.9%, 73.1%), while MicroscopyGPT performs well only in Tier 1 ($\approx$92%) but drops sharply in Tier 2–3. AtomAI, by contrast, stays below 2.7%. For structure success rate, AutoMat achieves 83.2% overall (85.0%, 84.0%, 73.1%), compared with only 25.9% for MicroscopyGPT in Tier 2 and almost zero for AtomAI.

Beyond comparisons with external baselines, we further compare AutoMat with a stronger same-tool heuristic script to test whether the controller contributes beyond a well-

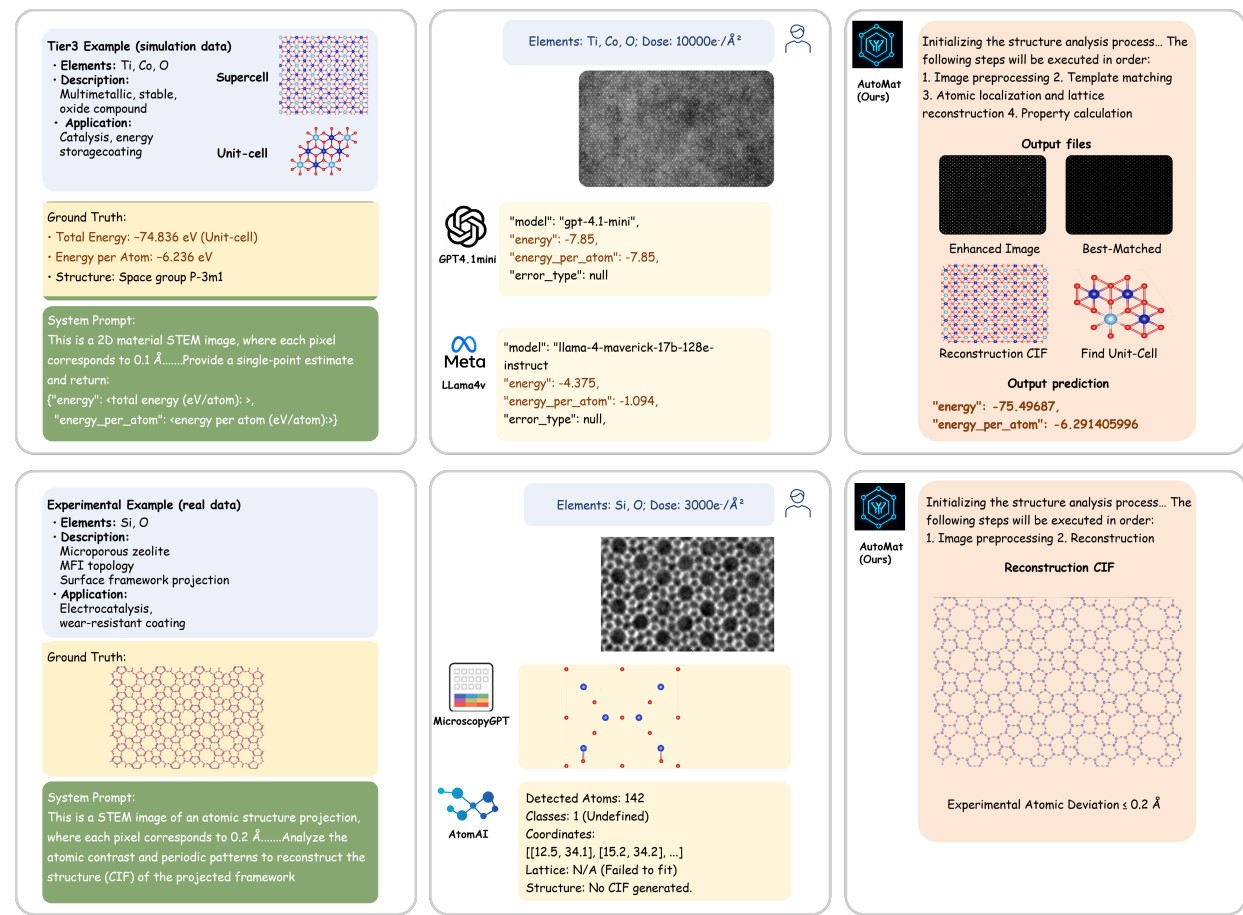

*Figure 4.* **Visualizing the reasoning gap.** We compare models on **(Top) a hard synthetic instance** and **(Bottom) a real-world ZSM-5 experiment**. While baselines suffer from *geometric hallucinations* or lack of structure due to missing physical constraints, **AutoMat** provides initial evidence of zero-shot transfer by orchestrating a physics-guided reasoning loop to recover plausible surface topologies.

engineered non-LLM pipeline. This heuristic script already uses the same scientific tools as AutoMat and includes hand-coded quality gating, retrieval branching, rollback, and bounded retries. As shown in Table 3, it provides a strong scripted baseline but remains consistently below AutoMat across energy and structural metrics. The gap is especially clear on Tier 3, where energy MAE decreases from 585.00 to 333.49 meV/atom and S.S. improves from 62.3% to 73.1%. Overall, AutoMat also improves Energy MAE from 385.42 to 321.57 meV/atom, C.C. from 76.5% to 83.2%, and S.S. from 74.2% to 83.2%. These results suggest that the gain of AutoMat is not merely due to the underlying tools or a few threshold rules, but to more flexible failure-aware orchestration and parameter adjustment under noisy, ambiguous, and multi-element reconstruction conditions.

In summary, **AutoMat** not only outperforms all baselines but also maintains strong results in challenging Tier 3 cases with multi-element compositions and low imaging doses,

demonstrating robustness and generalizability across the full benchmark.

### 5.2. Real iDPC-STEM Case Study and Human-Expert Effort

**Real iDPC-STEM case.** To complement the simulation-based benchmark, we also evaluate **AutoMat** on a real iDPC-STEM case: a ZSM-5 zeolite sample acquired on the same Cs-corrected STEM under dose-limited conditions. Without any manual tuning, the agent automatically performs denoising and structural reconstruction, and outputs a CIF structure whose projected lattice and channel framework qualitatively agree with the known crystallographic model. As shown in Figure 4 (bottom) and Appendix A.11, AutoMat provides preliminary evidence of transfer from 2D monolayer simulations to the 3D surface projection of the zeolite. This demonstrates that AutoMat has a certain ability to transfer the image–structure–property pipeline from abTEM simulations to real microscope images.

**Human-expert effort.** For comparison, we consulted a senior electron-microscopy expert in our group. For Tier-2-like STEM images, manually interpreting a $1024 \times 1024$ image, confirming lattice parameters and elemental species, and producing a simulation-ready CIF typically requires about 6–8 hours per sample. In contrast, on our benchmark hardware, **AutoMat** processes similar Tier-2 samples in roughly 2 minutes per case, yielding orders-of-magnitude higher throughput.

### 5.3. Error Analysis

To better understand the failure modes of **AutoMat**, we analyzed representative failure cases across all three tiers and identified two primary types of errors (a detailed analysis with examples is provided in Appendix A.10):

**(1) Template retrieval failure (39.3%).** In these cases, AutoMat failed to retrieve the correct structure from the template database, resulting in mismatches in atomic arrangements and element types. This led to cascading errors in structure, composition, and property predictions. Incorrect atom counts further caused large energy estimation errors.

**(2) Downstream failure despite correct template (60.7%).** Even with the correct template, downstream steps failed due to projection ambiguity or elemental confusion. In 40% of these cases, atoms appeared too close in the 2D projection, and the lack of z-axis information led to poor relaxation and inaccurate energy estimates. In 20.7%, elements with similar atomic numbers (e.g., C and O) exhibited indistinguishable contrast, causing misclassification and full breakdowns in lattice fitting and CIF generation.

These findings highlight two key directions for improving **AutoMat**: (i) improving retrieval robustness via uncertainty-aware methods; and (ii) overcoming 2D projection limits through 3D-aware modeling (Guo et al., 2023; Tang et al., 2024; 2025b) and enhanced modality integration. To make projection ambiguity visible to users, STEM2CIF also reports per-site confidence scores based on local fitting quality and nearest-neighbor geometry, and appends warning flags for low-confidence regions (see Appendix A.3). In addition, adaptive distribution alignment and graph-domain adaptation may help reduce the remaining gap between simulated STEM images and real experimental inputs (Chen et al., 2026). To further distinguish template-free reconstruction from unseen-family generalization, we provide a leave-one-chemical-family-out evaluation in Appendix A.4.

## 6. Conclusion

We proposed **AutoMat**, an end-to-end agent system that reconstructs material structures and predicts properties from STEM images. By integrating pattern-adaptive vision models, symmetry-constrained reconstruction, and LLM-driven orchestration, AutoMat achieves accurate alignment between microscopy data and atomic modeling, significantly outperforming existing methods in structural and energetic evaluation. Meanwhile, we propose STEM2Mat-Bench for this task. AutoMat advances autonomous materials research and AI-driven experimentation. Future work will focus on strengthening its role as a bridge between experimental characterization and theoretical computation.

## Acknowledgements

This work was supported by the National Natural Science Foundation of China (Nos. 22322203 and 22275110), the Tsinghua University Initiative Scientific Research Program, the Key Research and Development Program of Inner Mongolia and Ordos (Nos. Ordoslab-kjzc-202506 and Ordoslab-sysjc-202502), the HE Science Foundation, the Scientific Research Innovation Capability Support Project for Young Faculty (No. ZYGXQNJSKYCXNLZCXM-E7), and the Central Universities Young Faculty Research Innovation Capacity Support Program (No. SRICSPYF-ZY2025039).

## Impact Statement

This paper presents *AutoMat*, an agent that automates microscopy-to-structure reconstruction from iDPC-STEM images and enables downstream materials evaluation. The expected positive impact is to reduce manual workload and accelerate reproducible materials analysis.

Potential risks include incorrect reconstructions (e.g., under low-dose noise, similar-$Z$ elements, or projection ambiguity) being propagated into downstream simulations and property predictions, which could mislead scientific conclusions if used without verification. We therefore position AutoMat as an assistive tool and recommend inspecting intermediate outputs (retrieved templates, reconstructed structures, and logs) and applying expert validation, especially for out-of-distribution experimental conditions. AutoMat should not be used as a fully autonomous replacement for expert crystallographic analysis in high-stakes experimental studies; rather, it is intended to assist experts by accelerating candidate generation, consistency checking, and downstream validation.

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

# A. Appendix

## A.1. Dataset Overview

As illustrated in Figure 5, the atom count distribution (left) reveals a predominance of compact unit cells (median $N = 6.0$), ensuring computational feasibility for structural reasoning. The periodic table heatmap (right) demonstrates comprehensive elemental coverage spanning 67 unique species, with intensity peaks in chalcogens (e.g., O, S) and transition metals that accurately reflect the compositional diversity of stable 2D materials.

## A.2. MOE–DIVAESR: Training Pipeline & Architecture

### A. Training pipeline.

1. **Dataset partitioning.** A total of 1,693 denoised STEM images were projected to a PCA-reduced feature space and clustered using $k$-means ($k$=8). The resulting clusters were grouped into **2 parent** and **6 child** routing categories, defining the hierarchical MoE.

2. **Prototype selection & simulation.** For each child cluster, the top 25% most representative samples (424 total) were re-simulated into iDPC-STEM projections. These images were augmented with random rotation, flipping, Poisson noise, Gaussian blur, elastic deformation, and sliding-window cropping, resulting in a total of **756,042** training patches of size $128 \times 128$.

3. **Stage 1: Router pretraining.** A ResNet-18 model was trained as a routing network for 30 epochs to predict the appropriate expert index. It achieved a top-1 routing accuracy of **99%** on the validation set. The weights were then frozen for expert training.

4. **Stage 2: Expert training.** Six DIVAESR experts (EDSR backbone with 12 residual blocks and 64 feature maps) were trained jointly under fixed routing for 300 epochs. Compared to a single DIVAESR baseline, the MoE model reduced reconstruction error by approximately 50%.

### B. Model Architecture and Objective.

Each expert $E_k$ is modeled as a two-stage cascade of a denoising module and a super-resolution module:

$$E_k = g_{\phi_k}^{\text{SR}} \circ f_{\theta_k}^{\text{DIVAE}}, \qquad k = 1, \ldots, K. \tag{5}$$

where $f_{\theta_k}^{\text{DIVAE}}$ is the denoising network and $g_{\phi_k}^{\text{SR}}$ is the super-resolution network.

The router $R_\psi$ produces a one-hot routing mask $\mathbf{r} \in \{0,1\}^K$ that selects expert $k^*$ for each input:

$$k^* = R_\psi\left(x^{\text{noisy}}\right). \tag{6}$$

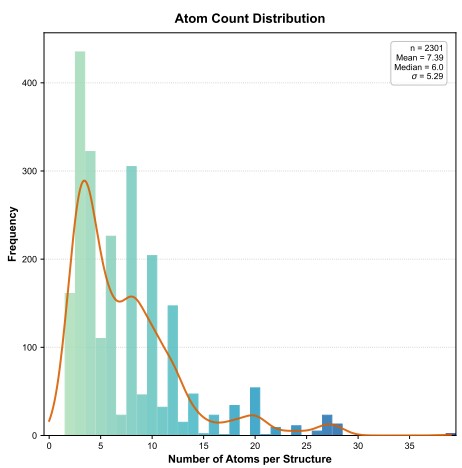
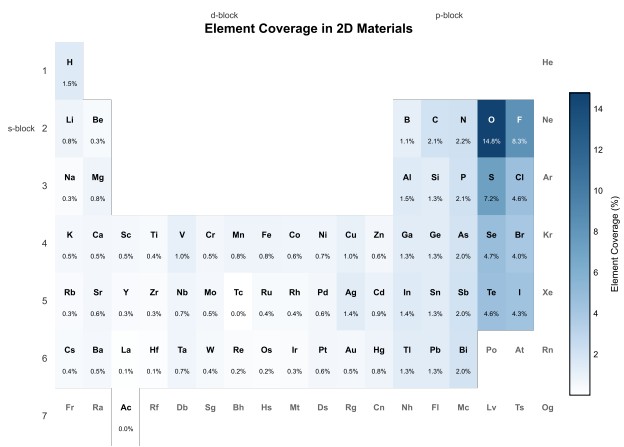

*Figure 5.* Distribution of per-unit-cell structure atom counts (left) and elemental coverage percentages in our curated 2D materials dataset (right).

The output of the MoE model is then computed as:

$$\hat{x}^{\text{HR}} = \sum_{k=1}^{K} r_k \cdot g_{\phi_k}^{\text{SR}} \left( f_{\theta_k}^{\text{DIVAE}}(x^{\text{noisy}}) \right). \qquad (7)$$

**Joint loss objective.** The total training loss for a mini-batch is given by:

$$\mathcal{L}_{\text{MoE}} = \sum_{k=1}^{K} r_k \cdot \left( \mathcal{L}_{\text{DIVAE}}^{(k)} + \mathcal{L}_{\text{SR}}^{(k)} \right) + \lambda \sum_{k=1}^{K} \bar{p}_k \log \bar{p}_k. \quad (8)$$

where

$$\mathcal{L}_{\text{DIVAE}}^{(k)} = \frac{1}{N} \sum_{i=1}^{N} \left\| \hat{x}_i^{\text{clean}} - x_i^{\text{clean}} \right\|_2^2 + \beta \cdot \text{KL} \left[ q(z|x_i^{\text{noisy}}) \, \| \, \mathcal{N}(0, I) \right] \quad (9)$$

is the $\beta$-VAE denoising loss, and

$$\mathcal{L}_{\text{SR}}^{(k)} = \frac{1}{N} \sum_{i=1}^{N} \left\| \hat{x}_i^{\text{HR}} - x_i^{\text{HR}} \right\|_1 \qquad (10)$$

is the L1 reconstruction loss for super-resolution. Here, $\bar{p}_k$ denotes the empirical routing frequency for expert $k$.

This formulation jointly minimizes noise and detail loss through specialized expert modules. The KL regularization in Eq. (9) stabilizes latent representations, while Eq. (10) preserves fine image details. The entropy term in Eq. (8) promotes balanced expert usage.

### A.3. Template Matching and STEM2CIF Conversion

### A. Image Template Matching.

1. **Feature extraction.** Atomic centroids $\{\mathbf{p}_i\}_{i=1}^{N}$ are detected in each denoised image. An $m{=}32$-bin radial-

distribution histogram $\mathbf{g} = (g_1, \ldots, g_m)$ is computed:

$$g_b = \frac{1}{N} \sum_{i<j} \mathbb{I}[r_{b-1} \leq \|\mathbf{p}_i - \mathbf{p}_j\| < r_b], \quad b = 1, \ldots, m. \qquad (11)$$

2. **Candidate retrieval.** All template images are similarly encoded as RDF descriptors $\{\mathbf{g}^{(t)}\}$. The cosine distance is computed as:

$$d_{\cos} \left( \mathbf{g}, \mathbf{g}^{(t)} \right) = 1 - \frac{\langle \mathbf{g}, \mathbf{g}^{(t)} \rangle}{\|\mathbf{g}\| \cdot \|\mathbf{g}^{(t)}\|}. \qquad (12)$$

The top-$k$ nearest templates ($k = 10$ by default) are retrieved.

3. **Element filtering.** Retain only templates whose elemental composition $\mathcal{E}^{(t)}$ matches the input:

$$\mathcal{E}^{(t)} = \mathcal{E}^{(\text{input})}. \qquad (13)$$

From the remaining candidates, select the one with the smallest $d_{\cos}$.

4. **Output.** The top-1 matched template is copied to the output directory for downstream reconstruction.

### B. STEM2CIF Conversion.

1. **Peak localization.** Atomic column centers are localized using weighted mean-shift clustering followed by sub-pixel 2D Gaussian fitting.

2. **Lattice fitting.** An initial lattice is fitted using least-squares optimization based on known pixel resolution (0.1 Å/pixel), constrained by the space-group symmetry of the matched template.

3. **Element assignment.** Atomic column intensities are compared to simulated iDPC-STEM contrast values, and the most probable element is assigned to each atomic site.

4. **CIF generation.** The reconstructed periodic structure is reduced to its primitive unit cell and exported as a standard CIF file for downstream MatterSim relaxation and property prediction.

## C. Confidence Estimation and Warning Flags.

To expose projection ambiguity to users, STEM2CIF additionally reports a per-site confidence score during reconstruction. The confidence is estimated from local lattice fitting quality and neighborhood geometry, using the local fitting residual, represented by the fitted $R^2$, together with the nearest neighbor (NN) distance. Intuitively, severely overlapping atomic columns or closely packed heavy atoms tend to produce lower confidence, while isolated well-fitted atoms yield higher confidence. When the confidence score falls below a predefined safe threshold, AutoMat appends a warning flag to the reconstructed output, indicating that the corresponding region should be inspected by an expert before downstream property analysis.

*Table 4.* **Examples of per-site confidence estimates from STEM2CIF.** Closely packed heavy atoms exhibit lower confidence, while well-separated atoms yield higher confidence.

| ID | Element | $x$ (Å) | $y$ (Å) | $R^2$ | NN Dist. (Å) | Confidence |
|----|---------|---------|---------|-------|--------------|------------|
| 65 | I | 12.85 | 65.10 | 0.972 | 1.48 | 0.744 |
| 53 | I | 13.29 | 62.46 | 0.969 | 1.67 | 0.654 |
| 21 | B | 40.54 | 35.65 | 0.977 | 6.22 | 0.953 |
| 41 | B | 68.28 | 53.80 | 0.993 | 2.91 | 0.876 |

As shown in Table 4, closely packed iodine columns receive lower confidence scores, whereas more isolated boron sites receive higher confidence scores. This mechanism does not fully resolve the single-projection depth ambiguity, but converts part of the physical ambiguity into a user-facing uncertainty signal.

## A.4. Ablation Study

The STEM2CIF module is the only component that converts microscopy images into atomic structures in the form of CIF files. Removing it would prevent structural outputs, making RMSD, composition correctness, and formation-energy MAE undefined. Therefore, ablating STEM2CIF is not meaningful.

To isolate contributions of other modules, we selectively disable (i) pattern-adaptive denoising (MOE-DIVAESR), and (ii) physics-guided template matching, while keeping the rest unchanged. We report formation-energy MAE in Table 5.

*Table 5.* Energy per-atom MAE (meV/atom) under different ablation settings. Lower is better.

| Method | Tier 1 | Tier 2 | Tier 3 |
|--------|--------|--------|--------|
| *No Denoising* | 6584 | 2616 | 938 |
| *No Template Retrieval* | 617 | 608 | 672 |
| **AutoMat (Full)** | **344** | **320** | **333** |

Removing MOE-DIVAESR significantly increases error in Tier 1, indicating its critical role in enhancing image quality. Removing template matching impacts Tier 3 most, where strong priors are essential under low-dose and multi-element conditions. These results suggest that template retrieval should not be interpreted as a purely marginal fallback. Rather, the current system is better viewed as combining two complementary reconstruction paths: a direct image-driven path via STEM2CIF, and a retrieval-guided path that provides a stronger structural prior when direct reconstruction becomes unreliable. The "No Template Retrieval" setting shows that the direct path remains operative, but the sharp degradation from 333 to 672 meV/atom on Tier 3 indicates that retrieval is especially important in hard low-dose, multi-element cases. We therefore revise the main text to describe retrieval as a *state-dependent auxiliary branch* rather than a purely secondary add-on.

## Leave-One-Chemical-Family-Out (LOFO) Evaluation.

To directly assess generalization beyond seen structural families, we conduct a leave-one-chemical-family-out (LOFO) evaluation. For each held-out family, we remove that family from both the training split and the retrieval library, and test only on that family. We report three settings: **Full AutoMat**, **LOFO-Full**, and **LOFO-NoTemplate**. The first measures the original in-family performance, the second measures unseen-family generalization under the full controller, and the third measures the standalone performance of the direct STEM2CIF path when no correct retrieval prior is available.

*Table 6.* Leave-one-chemical-family-out (LOFO) evaluation. Lower is better for RMSD$_{xy}$ and Energy MAE; higher is better for S.S. and C.C.

| Held-out family | Setting | RMSD$_{xy}$ (Å) ↓ | S.S. (%) ↑ | C.C. (%) ↑ | Energy MAE (meV/atom) ↓ |
|-----------------|---------|-------------------|------------|------------|--------------------------|
| MXenes ($N$=6) | Full AutoMat | 0.117 | 100.0 | 100.0 | 174 |
| | LOFO-Full | 0.122 | 100.0 | 97.2 | 186 |
| | LOFO-NoTemplate | 0.181 | 83.3 | 83.3 | 345 |
| Perovskites ($N$=53) | Full AutoMat | 0.124 | 76.7 | 82.1 | 356 |
| | LOFO-Full | 0.125 | 66.8 | 73.6 | 370 |
| | LOFO-NoTemplate | 0.183 | 54.9 | 60.4 | 685 |

The LOFO results help separate two questions that were previously conflated. First, **LOFO-Full** measures unseen-family generalization under the full controller. On MXenes, the drop from Full AutoMat to LOFO-Full is minimal, while on perovskites the degradation is moderate rather than catastrophic, indicating that AutoMat retains meaningful generalization beyond seen families. Second, **LOFO-NoTemplate**

measures how well the direct STEM2CIF path works when no correct retrieval prior is available. Here the performance drops much more substantially, especially for perovskites ($356 \rightarrow 685$ meV/atom), showing that the direct path remains operative but that the retrieval-guided branch is particularly important in structurally harder families. Overall, these results suggest that AutoMat is neither a pure retrieval system nor a fully retrieval-free one: it is better viewed as a dual-path framework combining direct reconstruction with a state-dependent retrieval-guided prior branch.

### A.5. Broader Impact

AutoMat can accelerate the discovery and validation of novel materials by automating structure reconstruction from electron microscopy images. This reduces reliance on expert annotation and lowers the barrier to entry in under-resourced research settings. Open release of data and code promotes transparency and reproducibility, fostering collaboration.

### A.6. Code and Data Availability

The code and dataset are publicly available at `https://github.com/yyt-2378/AutoMat` and `https://huggingface.co/datasets/yaotianvector/STEM2Mat`. The released repository includes the main AutoMat pipeline, evaluation scripts, and instructions for reproducing the benchmark experiments.

### A.7. Experimental Settings

**Framework.** Training and experiment management use `PyTorch Lightning` on a $4\times$ NVIDIA H100 NVLink server (80GB per GPU). Hyperparameter tuning for MOE-DIVAESR required approximately 4–5 days (see Table 7).

### A.8. MicroscopyGPT Baseline

MicroscopyGPT is used as an off-the-shelf vision–language baseline without additional fine-tuning. Following Choudhary *et al.*, MicroscopyGPT fine-tunes an 11B LLaMA-3.2–Vision model with QLoRA on simulated STEM images. Each example pairs a 2D STEM projection (mainly along (001)) with a structured textual description containing lattice parameters, lattice angles, element types, and fractional atomic coordinates.

We do *not* further fine-tune MicroscopyGPT on STEM2Mat-Bench to avoid overfitting and to keep baselines comparable. For each benchmark image, we (i) feed the STEM image with a standard structure-generation prompt, (ii) parse the textual output into a structured representation, and (iii) convert it into a CIF file using ASE utilities.

*Table 7.* Configuration of MOE-DIVAESR and related submodules.

| Gating Network | |
| --- | --- |
| Parent Router Modules | 2 |
| Child Router Modules | 6 |
| **Latent Prior Module (DIVAESR)** | |
| Input Channels | 1 |
| Latent Dimension | 128 |
| **Super-Resolution Module (DIVAESR)** | |
| Residual Blocks | 12 |
| Feature Maps per Block | 64 |
| Activation Function | ReLU |
| Patch Size | 128 |
| Scale Factor | $\times 2$ |
| Image Channels | 1 (grayscale) |
| Precision | FP32 |
| **Optimization and Training Settings** | |
| Optimizer | Adam |
| Initial Learning Rate | 0.001 |
| Weight Decay | 0.0 |
| KL Divergence Weight | 0.00025 |
| Training Epochs | 300 |
| Training Batch Size | 128 |

### A.9. Robustness Across Agents and Retry

**Retry Mechanism.** We implement **in-context learning from environmental feedback**, where tool error messages are fed back into subsequent turns. This enables the agent to refine calls until a maximum retry limit is reached or the tool call succeeds. This retry-and-repair mechanism is motivated by the broader need to improve LLM resilience under erroneous reasoning and perturbed execution states.

```
[TOOL_CALL] reconstruction_tool
{"img_path": ...,
"elements": ["Co", "Fe", "O"]}}

[TOOL_RESPONSE] reconstruction_tool
ValueError: Unexpected "}"

[ANSWER]
Malformed arguments detected.
Correcting and retrying.

[TOOL_CALL] reconstruction_tool
{"img_path": ...,
"elements": ["Co", "Fe", "O"]}

[TOOL_RESPONSE] reconstruction_tool
{success: true, cif_path: ...}

[TOOL_CALL] validation_tool
{"cif_path": ...}

[TOOL_RESPONSE] validation_tool
```

```
{converged: true, stress_xx_GPa: -10.956}

[ANSWER]
High residual stress detected.
Returning a low-confidence warning.
```

**State-Dependent Tool Invocation.** AutoMat does not execute tools in a fixed order. Instead, tool invocation is conditioned on intermediate states such as denoising quality and reconstruction validity. In particular, template retrieval is not always required: when the denoised image is sufficiently reliable, the agent can directly invoke STEM2CIF and skip retrieval; retrieval is only triggered when the direct path fails or is judged unreliable.

```
[TOOL_CALL] denoise_tool
{"image_path": ...}

[TOOL_RESPONSE] denoise_tool
{success: true, recon_img: ...}

[ANSWER]
Denoising completed.
The reconstructed image appears
sufficiently clean for direct reconstruction,
so I will first attempt STEM2CIF
without template retrieval.

[TOOL_CALL] stem2cif_tool
{"img_path": ..., "elements": [...]}

[TOOL_RESPONSE] stem2cif_tool
{success: true, cif_path: ...}

[TOOL_CALL] validation_tool
{"cif_path": ...}

[TOOL_RESPONSE] validation_tool
{success: true,
converged: true, stress_xx_GPa: ...}

[ANSWER]
The direct path succeeded.
Template retrieval was not
invoked in this case.
```

**Overall sensitivity to the choice of LLM/VLM agent.** We replace the agent backbone with Qwen3, GPT-4o, and Qwen-VL (72B) and evaluate on a stratified random sample of 230 cases. Performance remains stable, indicating that gains stem from the framework rather than a particular backbone.

**Decision-making flexibility.** AutoMat is an agentic controller rather than a fixed pipeline: it dynamically decomposes subtasks and adapts tool invocation. If a tool fails, rollback-and-retry is triggered until success or termination.

*Table 8.* Formation-energy MAE (meV/atom) across tiers for different agents.

| Method | Tier 1 | Tier 2 | Tier 3 | Avg. |
|---|---|---|---|---|
| Qwen-VL (72B) | 261.6 | 308.5 | 446.1 | 313.8 |
| Qwen3 agent | 245.2 | 306.2 | 435.1 | 310.5 |
| GPT-4o agent | 240.7 | 304.2 | 435.0 | 308.5 |
| DeepSeekV3 | 243.8 | 305.3 | 433.5 | 309.6 |

*Table 9.* Structural accuracy metrics across tiers.

| Tier | Method | RMSD (Å) | C.C. (%) | S.S. (%) |
|---|---|---|---|---|
| 1 | Qwen-VL | $0.12 \pm 0.02$ | 100 | 100 |
| | DeepSeekV3 | $0.11 \pm 0.02$ | 100 | 100 |
| 2 | Qwen-VL | $0.11 \pm 0.03$ | 83.1 | 79.2 |
| | DeepSeekV3 | $0.11 \pm 0.02$ | 85.9 | 79.8 |
| 3 | Qwen-VL | $0.12 \pm 0.02$ | 58.3 | 58.3 |
| | DeepSeekV3 | $0.11 \pm 0.03$ | 66.7 | 66.7 |

### A.10. Failure Analysis

We categorize failure cases into:

- **Template-retrieval failures (39.3%).** The correct structure is absent from the template database. *Representative case:* Tier 1, `2dm-5936` (Se-oxide vs. nitride). High-$Z$ Se columns resemble N-like contrast.

- **Downstream failures (60.7%).** The correct template is retrieved, but downstream reconstruction fails due to projection effects or extreme contrast imbalance. See Table 11.

### A.11. Real STEM Image Reconstruction on ZSM-5

Due to experimental limitations, our current setup does not yet support systematic preparation and iDPC-STEM imaging of 2D materials. To provide an informative evaluation on real experimental data, we choose ZSM-5 zeolite as a representative test case.

**ZSM-5 iDPC-STEM imaging experiments.** Experiments were carried out on a Cs-corrected STEM (FEI Titan Cubed Themis G2 300) equipped with a four-quadrant detector at 300 kV. Sheet-like ZSM-5 single crystals were ultrasonically dispersed in pyridine or thiophene for 1–2 h, deposited onto microgrids, and heated at 130 °C.

**AutoMat reconstruction on real ZSM-5 images.** AutoMat recovers the framework topology and continuous ten-membered-ring channels consistent with the reference. After alignment, framework atomic deviations are within $\sim 0.2$ Å, indicating reasonable structural fidelity under realistic experimental noise.

*Table 10.* Tool invocation, success, and retry statistics of different agents in the AutoMat toolchain.

| LLM Agent | Tool | Total Calls | Callbacks | Successes | Retries | Success Rate (%) |
|---|---|---|---|---|---|---|
| Qwen-VL (72B) | Denoising | 293,417 | 259,217 | 259,217 | 34,200 | 88.34 |
| | Property Prediction | 165,594 | 121,153 | 68,532 | 97,062 | 41.39 |
| | STEM2CIF | 314,001 | 252,801 | 133,282 | 180,719 | 42.45 |
| | Template Matching | 258,987 | 205,762 | 205,762 | 53,225 | 79.45 |
| GPT-4o | Denoising | 58,053 | 56,950 | 56,950 | 1,103 | 98.10 |
| | Property Prediction | 38,705 | 36,563 | 36,563 | 2,142 | 94.46 |
| | STEM2CIF | 46,600 | 43,507 | 43,507 | 3,093 | 93.36 |
| | Template Matching | 59,506 | 55,951 | 55,951 | 3,555 | 94.02 |

*Table 11.* Breakdown of downstream failure sub-modes despite correct template retrieval.

| Sub-mode | Share | Core Problem | Representative Case |
|---|---|---|---|
| Projection overlap | 40% | Atoms in different $z$-layers project to the same $(x, y)$, confusing structural relaxation | Tier 2, `2dm-1014`: B/I overlap; I dominates iDPC, B is lost |
| Extreme $Z$-contrast | 20.7% | Heavy elements dominate contrast and mask lighter atoms, treated as noise | Tier 3, `2dm-5199`: U contrast hides O/F, leading to $> 3$ eV energy error |

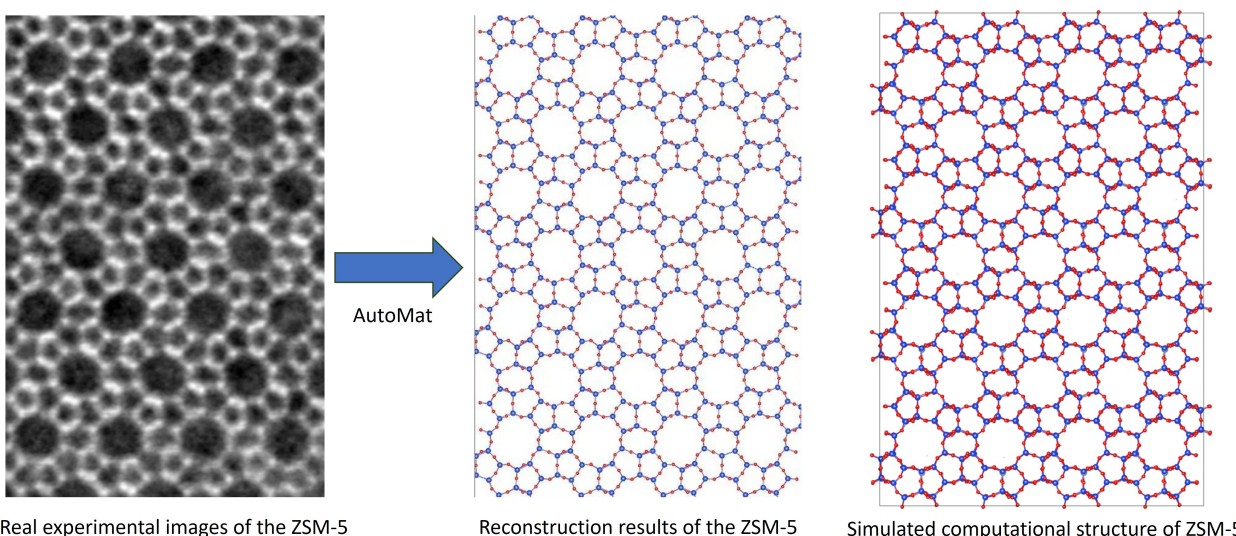

Real experimental images of the ZSM-5     Reconstruction results of the ZSM-5     Simulated computational structure of ZSM-5

*Figure 6.* **AutoMat reconstruction on real iDPC-STEM images of ZSM-5 zeolite.** (Left) Experimental iDPC-STEM image showing characteristic ten-membered-ring pores. (Middle) Structure reconstructed by AutoMat from the real image. (Right) DFT-relaxed reference model of ZSM-5.

### A.12. Limitations and Future Directions

AutoMat currently targets **2D monolayer systems imaged with iDPC-STEM**, which provides a stable and interpretable testbed but also introduces several important limitations.

- **Limited real-data validation.** Although we include a real iDPC-STEM case on ZSM-5 in Appendix A.11, this experiment should be interpreted as an *initial feasibility demonstration* rather than a substitute for systematic validation on multiple real 2D monolayer materials. High-quality raw iDPC-STEM data for 2D materials are cur-

rently scarce in public repositories, and acquiring new large-field, dose-limited experimental data requires substantial sample-preparation effort, aberration-corrected STEM beamtime, and repeated calibration. Broader real-material validation therefore remains an important next step.

- **2D scope and single-projection depth ambiguity.** The current pipeline assumes a single iDPC-STEM projection. Since iDPC-STEM mainly reflects the projected electrostatic potential integrated along the beam direction, a single image does not preserve sufficient depth information to uniquely recover a general 3D atomic structure.

This makes reconstruction of bulk 3D crystals intrinsically ill-posed from a single projection. By contrast, 2D monolayer materials have limited extent along the $z$ axis, which substantially reduces projection ambiguity and makes them a physically well-posed starting point.

- **Element classification uncertainty.** Element assignment relies on contrast cues and template priors; similar-$Z$ elements and low-dose conditions increase confusion.

- **Template retrieval sensitivity.** Mismatches in pixel scale, defocus, and aberrations reduce robustness.

- **Heuristic agent policy.** The current policy uses state-dependent rules; it may not remain optimal as the tool pool grows.

**Future directions.** We plan to address these limitations along four main directions. First, for **real-data robustness**, we will expand real-image evaluation to multiple monolayer materials with known CIFs, enlarge the template library to cover a broader range of defocus and aberration conditions, and introduce simple calibration steps for pixel scale and imaging-condition drift. Second, for **3D extension**, we are exploring two concrete routes: *electron tomography*, where multiple tilt-angle projections are combined for 3D reconstruction, and *4D-STEM*, where diffraction information at each probe position may provide richer depth-sensitive structural signals. Third, we plan to adopt **contrast-sensitive recognition** and fuse complementary modalities such as **EELS/EDX** to improve species discrimination under overlap or similar-$Z$ conditions. Finally, we will explore **learnable policy optimization** and more robust retrieval strategies, including **pre-retrieval calibration**, to improve controller scalability and robustness under broader experimental conditions.

