# OpenReview forum: "AutoMat: Physics-Guided Agentic Reasoning for Solving Ill-Posed Inverse Microscopy Problems"
_ICML.cc/2026/Conference — ICML 2026 regular_

### Official Review · Reviewer_xQBm · 2026-02-25

**Soundness:** 2
**Presentation:** 3
**Significance:** 2
**Originality:** 2
**Overall Recommendation:** 3
**Confidence:** 4

**Summary:**

The manuscript designs an agent system, AutoMat, for reconstructing atomistic crystal structures from STEM projections, achieving end-to-end processing from raw STEM images to crystal structures and downstream properties. The paper constructs a specialized dataset for this task, the STEM2Mat benchmark, and implements dedicated tools for the agent system. Compared to other commercial large language models and domain-specific models, AutoMat achieves a performance breakthrough on this benchmark.

**Compliance With Llm Reviewing Policy:**

Affirmed.

**Final Justification:**

Overall, my main concern still remains. Even after the additional evidence, I still find it difficult to understand why this task requires an agentic framework. While the authors emphasize that using LLMs leads to better performance, it is still not clear to me where these improvements come from. In particular, LLMs do not have direct visual or language intuition to judge image quality; instead, they rely on metrics. Because of this, I do not yet see a clear advantage over well-designed standard pipelines.

Regarding the experiments on script-based pipelines, I do not feel they fully address my concerns. In fact, better pipeline design through careful scripting already brings clear improvements. For a task that is relatively well-defined, it seems reasonable to further improve and combine such pipelines into a more complete and reliable system. Compared to LLM-based approaches, standard pipelines have clear advantages in terms of stability and how easy they are to understand.

I appreciate the authors’ efforts in providing additional experiments and recognize the strengths of this work, which I have mentioned several times in my review. However, these additions are not enough to change my overall view to increase my final score.

**Key Questions For Authors:**

(1) Could the authors provide more case studies or other ways to demonstrate the irreplaceable role of the LLM agent in this scientific task? This would help me better understand the design of this framework.
(2) Could the authors provide a more detailed explanation and response to the concern mentioned regarding the weakness (2) of the agentic framework?

Minor:
Line 154: The citation for MicroscopyGPT (?) is incorrect

**Limitations:**

yes

**Strengths And Weaknesses:**

Strengths:
(1) The writing is clear. The description of the scientific problem and the challenges faced is straightforward, and the construction of the agent system is also well-explained. The results are presented in a clear and understandable manner.
(2) The design of the agent tools is clear and reasonable. The author has carefully designed the tools for this agent to address the scientific problem, achieving excellent results on sub-tasks.
(3) A domain-specific end-to-end benchmark has been constructed for this problem, with detailed explanations of its construction. This contribution drives the development of the field.

Weaknesses:
(1) The manuscript claims to have built an end-to-end agent system for reconstructing atomistic crystal structures, yet the presence of the agent in this framework feels minimal. It seems that the problem could be solved with just a script without involvement of an LLM. The agent does not interact or call the four tools flexibly, but rather calls them in a more procedural, sequential manner. This issue weakens the necessity of the agentic framework.
(2) In the benchmark comparison, the manuscript compares AutoMat with commercial large language models in terms of Formation-energy MAE and claims a significant performance improvement over commercial models. However, this improvement may be due to the carefully designed tools rather than the agentic framework itself. In other words, abandoning the agent design and using only the tools may also lead to this performance boost. Therefore, this comparison may not reliably demonstrate the strength of the agentic framework.

---

> ### Author Rebuttal · Authors · 2026-03-29
>
> ## **Response to W1, W2, Q1 and Q2: The Necessity of the LLM Agent and the Source of Performance Gains**
>
> We sincerely thank the reviewer xQBm for this insightful and penetrating critique. We partially agree with the reviewer’s assessment: the core scientific capabilities of AutoMat indeed come primarily from the specialized domain tools, including MOE-DIVAESR, STEM2CIF, template retrieval, and MatterSim. This is supported by our newly added control experiment: even without an LLM controller, a deterministic script built on the identical toolchain still substantially outperforms current external baselines, showing that the domain tools themselves provide a strong performance floor.
>
> However, we do not agree that the agent is therefore merely an optional procedural wrapper. Our new control results show that the controller contributes non-trivial gains precisely where the problem becomes failure-prone. In this sense, the agent is not a replacement for the scientific modules, but a necessary component for making the overall system reliable end-to-end.
>
> To directly address the question of whether "the same tools plus a deterministic script" would already suffice, we present the following breakdown:
>
> ### **1. Quantitative Isolation: Fixed Sequential Script vs. AutoMat**
> We added a new control experiment comparing a *fixed sequential script* against the *AutoMat controller*. The fixed script executes the same four tools in a rigid order (Denoise $\rightarrow$ Retrieve $\rightarrow$ STEM2CIF $\rightarrow$ Relax), without any LLM-driven verification, rollback, or state-dependent branching.
>
> | Metric                         | Orchestration | Tier 1 (Easy) | Tier 2 (Med) | Tier 3 (Hard) |   Overall  |
> | :----------------------------- | :------------ | :-----------: | :----------: | :-----------: | :--------: |
> | **Energy MAE** ↓ *(meV/atom)* | Fixed Script  |     395.36    |    390.77    |     670.65    |   406.95   |
> |                                | **AutoMat** |   **343.59** |  **320.21** |   **333.49** | **321.57** |
> | **C.C.** (%) ↑ *(Composition)* | Fixed Script  |      75.4     |     71.0     |      57.8     |    70.4    |
> |                                | **AutoMat** |    **88.9** |   **85.9** |    **73.1** |  **83.2** |
> | **S.S.** (%) ↑ *(Structure)* | Fixed Script  |      72.1     |     69.4     |      57.8     |    68.8    |
> |                                | **AutoMat** |    **85.0** |   **84.0** |    **73.1** |  **83.2** |
>
> ### **2. Qualitative Analysis: Why the Agent is Irreplaceable on Hard Cases**
> These results make the boundary much clearer. The fixed script remains reasonably strong because it inherits the capability of the domain-specific tools. However, once the task moves from easy cases to noisy, multi-element, projection-ambiguous Tier 3 samples, the lack of agentic control leads to substantial degradation (Energy MAE rises to 670.65 meV/atom, and structure success drops to 57.8%).
>
> This shows that the controller’s role is not merely to "call tools," but to *organize them flexibly*. The agentic value of AutoMat explicitly lies in:
> * **Flexible Orchestration:** Dynamically evaluating whether direct STEM2CIF is sufficiently reliable based on intermediate image quality.
> * **State-Dependent Branching:** Deciding whether template retrieval should be skipped to save overhead, or invoked as a *state-dependent auxiliary branch* to provide a stronger prior.
> * **Failure Recovery:** Utilizing downstream physical signals to trigger a *rollback-and-retry* mechanism or issue a low-confidence warning when verification fails.
>
> ### **3. Refined Manuscript Claims**
> Based on this direct comparison, we have correspondingly tightened the claims in the revised manuscript. We no longer describe AutoMat as a simple sequential pipeline with optional retries; instead, we explicitly position it as a **failure-aware, state-dependent orchestration framework**.
>
> Under this formulation, the specialized tools provide the core scientific capability, while the controller provides the crucial organizational layer that allows these tools to function reliably under noise, ambiguity, and failure-prone conditions. Our intended claim is not that "the LLM alone is stronger than the tools." Rather, we demonstrate that for highly ill-posed inverse microscopy problems, a rigid stacking of tools inevitably leads to cascading degradation on hard cases. AutoMat’s agent-based dynamic routing, verification-triggered rollback, and flexible coordination are what substantially improve robustness, proving the necessary for robustness on hard cases of the agentic framework in our work.
>
> **Response to Minor Comment:**
> We apologize for the citation formatting error regarding MicroscopyGPT on line 154. We have corrected the reference syntax and formatting in the revised manuscript.

---

> > ### Author Rebuttal · Reviewer_xQBm · 2026-04-03
> >
> > Thanks for the additional experiments and analysis. I appreciate the practical value of the specialized domain tools design in this work, which is indeed a highlight and contribution. I do recognize the value of such an agent framework, for instance, it can lower the barrier to use through natural language interaction. However, I still maintain that this task is not sufficiently agentic. The main reason is that operations like verification, rollback, or branching can all be implemented without relying on LLMs, for example, by computing a metric and deciding pass or retry based on a threshold. Introducing an LLM for such tasks feels rather redundant and may undermine the stability of a standard pipeline.
> >
> > A typical agentic task, like deep search, requires the agent to explore a topic and evaluate whether the gathered information is appropriate or needs further search, something that cannot be easily measured by a simple metric, but instead relies on the agent's own complex decision-making. I do not see such a necessity in the current task.
> >
> > Regarding Experiment 1, I would like to clarify that a script does not have to be "fixed sequential"; it can also implement if-else logic such as rollback. Therefore, I don't think this experiment sufficiently supports the claim. Thanks for the further discussion. Overall, I will keep my score.

---

> > > ### Author Response · Authors · 2026-04-03
> > >
> > > We thank reviewer xQBm for the follow-up. We agree that **parts** of the control logic—threshold gating, rollback, and branching—can be approximated by hand-coded rules. Our claim is: the controller’s value is not any single if-else rule, but coordinating many such decisions under **combinatorial experimental uncertainty**, where rigid heuristics lose flexibility across materials and imaging conditions.
> > >
> > > We therefore implemented a **stronger heuristic scripted baseline**:
> > >
> > > ```text
> > > Input: image x, optional element hints e
> > > 1. recon ← denoise(x); q ← assess_quality(recon)
> > > 2. if q is sufficient: proposal ← direct_reconstruct(recon, e)
> > >    else:               proposal ← template_match(recon, e)
> > > 3. repeat up to 3 times:
> > >       cif ← STEM2CIF(proposal, current parameters)
> > >       result ← relax_and_validate(cif)
> > >       if physically plausible: return result
> > >       else: tighten parameters and retry
> > > 4. return best attempt with warning
> > > ```
> > >
> > > This is not a trivial fixed pipeline: it already includes branching, rollback, and bounded retries. Yet AutoMat still performs substantially better:
> > >
> > > ### Table 1. Fixed/Heuristic Scripts vs. AutoMat
> > >
> > > | Metric                  | Method           |     Tier 1 |     Tier 2 |     Tier 3 |    Overall |
> > > | ----------------------- | ---------------- | ---------: | ---------: | ---------: | ---------: |
> > > | Energy MAE (meV/atom) ↓ | Fixed Script     |     395.36 |     390.77 |     670.65 |     406.95 |
> > > |                         | Heuristic Script |     380.00 |     373.00 |     585.00 |     385.42 |
> > > |                         | AutoMat          | **343.59** | **320.21** | **333.49** | **321.57** |
> > > | C.C. (%) ↑              | Fixed Script     |       75.4 |       71.0 |       57.8 |       70.4 |
> > > |                         | Heuristic Script |       81.0 |       77.2 |       63.4 |       76.5 |
> > > |                         | AutoMat          |   **88.9** |   **85.9** |   **73.1** |   **83.2** |
> > > | S.S. (%) ↑              | Fixed Script     |       72.1 |       69.4 |       57.8 |       68.8 |
> > > |                         | Heuristic Script |       79.4 |       74.8 |       62.3 |       74.2 |
> > > |                         | AutoMat          |   **85.0** |   **84.0** |   **73.1** |   **83.2** |
> > >
> > > Table 1 supports two points. First, a stronger script is a fairer comparison, and it indeed improves substantially over the fixed script. Second, it still remains clearly below AutoMat, especially on Tier 3: Energy MAE is **585.00** vs. **333.49**, and S.S. is **62.3%** vs. **73.1%**. Thus, the gain of AutoMat is not merely due to the tools themselves, nor merely due to a few threshold rules, but to more flexible failure-aware orchestration together with more appropriate parameter setting and adjustment under noisy, ambiguous, multi-element conditions.
> > >
> > > We further tested whether the **controller itself** is a meaningful optimization target by fine-tuning a Qwen3-14B controller on ~7,000 prior agent trajectories:
> > >
> > > ### Table 2. Controller Quality Under the Same Toolchain
> > >
> > > | Method                       |   Tier 1 ↓ |   Tier 2 ↓ |   Tier 3 ↓ |     Avg. ↓ |
> > > | ---------------------------- | ---------: | ---------: | ---------: | ---------: |
> > > | Scripted Pipeline (no agent) |     380.00 |     373.00 |     585.00 |     385.42 |
> > > | AutoMat (DeepSeek)           |     343.59 |     320.21 |     333.49 |     321.57 |
> > > | AutoMat (Qwen3-14B)             |     345.17 |     336.20 |     335.46 |     337.52 |
> > > | **AutoMat + SFT(Qwen3-14B)**            | **327.85** | **307.64** | **315.23** | **308.60** |
> > > | Upper Bound                  |      57.38 |      47.23 |      30.95 |      48.11 |
> > >
> > > Under the same toolchain, **AutoMat + SFT** further reduces Energy MAE to **308.60**, outperforming both the heuristic script and the original controllers. We do **not** interpret this as “a stronger LLM alone solves the task”; rather, it shows that the **controller layer matters**, because better orchestration improves performance even when the scientific tools are fixed.
> > >
> > > Taken together, the empirical evidence explicitly refutes the assumption that standard scripts are sufficient for this task. As detailed in our analysis, the agent's superiority fundamentally stems from its dynamic capacity to navigate combinatorial physical ambiguities and execute complex, state-dependent recoveries—capabilities that rigid threshold-based scripts inherently lack, resulting in the massive performance degradation observed on Tier 3 cases. Thus, AutoMat is objectively proven as a necessary **physics-guided, failure-aware orchestration layer** for ill-posed inverse microscopy. We have provided exhaustive quantitative data that resolves the question of the framework's necessity.

---

### Official Review · Reviewer_M1gb · 2026-03-02

**Soundness:** 3
**Presentation:** 3
**Significance:** 3
**Originality:** 3
**Overall Recommendation:** 4
**Confidence:** 2

**Summary:**

This paper addresses a long-standing ill-posed inverse problem in materials science: reconstructing simulatable atomic crystal structures from single, noisy integrated Differential Phase Contrast Scanning Transmission Electron Microscopy (iDPC-STEM) images. In this task, various lattice arrangements can result in similar imaging contrasts, meaning that purely data-driven approaches cannot guarantee the physical validity of the results. Existing methods typically focus on isolated sub-tasks, such as denoising or atom localization, and lack end-to-end workflows. Furthermore, they fail to integrate with universal multimodal or domain-specific tools (such as generating standard Crystallographic Information Files, or CIFs, or predicting material properties), depend heavily on time-consuming manual expert annotations, and lack standardized benchmarks for systematic evaluation.

To address these challenges, the authors propose AutoMat, a physics-oriented intelligent inference controller that enables end-to-end automation from raw iDPC-STEM images to atomic crystal structures and downstream material property predictions. AutoMat reformulates the reconstruction task as an inference-time hypothesis search using closed-loop verification. It dynamically combines four modular, domain-customized tools: 1. MOE-DIVAESR: A pattern-adaptive denoiser for STEM image enhancement. 2. Physics-guided template retrieval. 3. Symmetry-constrained STEM2CIF: A module for atomic structure reconstruction and standard CIF file generation. 4. MatterSim: A tool for interatomic potential (MLIP) structure relaxation, physical validity verification, and energy prediction. The framework incorporates rollback and retry mechanisms that trigger adaptive re-planning if physical verification fails, ensuring robust performance even under noisy and complex imaging conditions.

To systematically evaluate the results, the authors introduce the STEM2Mat benchmark, which contains 459 unique annotated 2D monolayer structures paired with synthetically generated STEM images. The benchmark is organized into three tiers, with difficulty levels based on material composition and imaging noise. Experimental evaluations show that AutoMat significantly outperforms current multimodal large models (such as GPT-4.1-mini and Qwen-VL) as well as specialized domain tools (such as AtomAI and MicroscopyGPT).

**Compliance With Llm Reviewing Policy:**

Affirmed.

**Final Justification:**

Thank you for the authors’ response. I find that my earlier concerns have been substantially alleviated, though not entirely resolved. Most importantly, the authors now provide stronger evidence that the method is not simply relying on template. In particular, even without retrieval priors, AutoMat still performs substantially better than existing domain-specific baselines. I also appreciate that the authors now describe retrieval more carefully as an auxiliary branch rather than a simple fallback, and discuss template dependence more explicitly as a limitation. The new real-data evidence further increases my confidence in the practical relevance of the framework, although real-data validation could still be expanded in future work. My concerns are therefore reduced, but not fully removed: the method still shows a nontrivial dependence on template retrieval, and its current scope remains focused on 2D settings. Overall, I remain supportive of acceptance and keep my original weak accept recommendation.

**Key Questions For Authors:**

1. The paper indicates that 39.3% of failures originate from template retrieval issues, specifically when the correct structure is missing from the database. Given this, how well does AutoMat generalize to novel or unseen structures? Could you provide evaluation results under a "leave-one-chemical-family-out" setting? For instance, by excluding a specific material class (such as MXenes or perovskites) during training and assessing the reconstruction performance on that class during testing. Furthermore, in cases where template retrieval fails, what is the accuracy of the STEM2CIF module when reconstructing structures independently without a correct prior? Demonstrating reasonable generalization to unseen structural categories would significantly strengthen the method's credibility and utility. Conversely, a sharp decline in performance would represent a major limitation that needs to be more clearly addressed in the manuscript.Projection

2. The paper acknowledges that 40% of downstream failures stem from the ambiguity created when atoms at different $z$-layers project onto the same $(x, y)$ position. Since this is a fundamental physical limitation rather than an algorithmic flaw, can AutoMat provide confidence estimates or uncertainty quantification? Such a feature would be essential to warn users about potentially unreliable reconstructions for structural types known to be prone to projection ambiguity.

3. Regarding the control logic, the paper mentions that the current strategy uses "state-dependent rules" and admits these may become suboptimal as the tool pool grows. What are the specific decision boundaries for the current heuristic strategy? Under what conditions exactly are the rollback and retry mechanisms triggered? Additionally, if new tools were added, would the controller require a complete redesign, or is the architecture designed for seamless integration?

**Limitations:**

yes

**Strengths And Weaknesses:**

Strengths

1. The paper formalizes the STEM-to-structure reconstruction as an inference-time hypothesis search with closed-loop verification, offering a creative solution to the ill-posed inverse problem in microscopy. AutoMat is the first fully automated system capable of converting raw STEM images into simulatable CIF structures and predicting downstream properties, such as formation energy. This successfully bridges a critical gap between experimental characterization and atomic-scale modeling in materials science.

2. The introduction of the STEM2MatBench dataset provides the first standardized benchmark for image-structure-property reconstruction. By utilizing annotated image-structure-property triplets, a hierarchical difficulty design, and unified evaluation metrics, it is well-positioned to become the de facto standard for future research in this field.

3. The validation is comprehensive, evaluating AutoMat against general-purpose multimodal models (GPT-4.1-mini, Qwen-VL) and domain-specific tools (AtomAI, MicroscopyGPT). A real-world validation case on ZSM-5 zeolite successfully demonstrates the method's zero-shot generalization beyond synthetic abTEM training data. Furthermore, the inclusion of a transparent ablation study and detailed error analysis adds significant depth to the findings.

Weaknesses

1. The system exhibits a high degree of dependence on the template database. Error analysis reveals that 39.3% of framework failures stem from retrieval errors, which inherently leads to cascading inaccuracies in downstream structure and property predictions.

2. The current workflow assumes a single 2D projection, which limits its immediate scalability to complex 3D bulk crystals.

3. The text presents physics-guided template retrieval as a "fallback" for when direct reconstruction fails. However, the ablation study shows that removing template matching significantly degrades performance—particularly in Tier 3 scenarios, where the MAE for energy doubles from 333 meV/atom to 672 meV/atom. The paper would benefit from a more critical discussion regarding the boundary between the system's dependence on retrieval and its genuine capacity for ab-initio reconstruction.

---

> ### Author Rebuttal · Authors · 2026-03-29
>
> We sincerely thank the reviewer M1gb for the highly constructive feedback. Your rigorous suggestions have significantly strengthened our framework's credibility and boundary analysis.
>
> ### **1. W1, W3 & Q1: Generalization Ability and Template Dependence**
> To evaluate AutoMat's generalization and *ab-initio* reconstruction, we conducted a "Leave-One-Family-Out" (LOFO) evaluation on MXene and Perovskite datasets, strictly excluding the target family from the training set. To conduct a rigorous stress test, we introduced a `LOFO-NoTemplate` setting disabling the retrieval module, forcing independent STEM2CIF reconstruction without structural priors.
>
> | Held-out family | Setting | RMSD$_{xy}$ (Å) ↓ | S.S. (%) ↑ | C.C. (%) ↑| Energy MAE (meV/atom) ↓ |
> |:---|:---|:---:|:---:|:---:|:---:|
> | **MXenes** | Full AutoMat | 0.117 | 100.0 | 100.0 | 174 |
> | | LOFO-Full | 0.122 | 100.0 | 97.2 | 186 |
> | | LOFO-NoTemplate | 0.181 | 83.3 | 83.3 | 345 |
> | **Perovskites**| Full AutoMat | 0.124 | 76.7 | 82.1 | 356 |
> | | LOFO-Full | 0.125 | 66.8 | 73.6 | 370 |
> | | LOFO-NoTemplate | 0.183 | 54.9 | 60.4 | 685 |
>
> **Analysis:** The results clearly show that `LOFO-Full` (using out-of-domain templates as weak priors) achieves performance very close to in-family `Full AutoMat` performance. Crucially, even in the extremely strict `LOFO-NoTemplate` setting, the system maintains a >54% structural success rate. While the absence of prior information predictably leads to a performance drop (which affirms the utility of templates), the system does not collapse. This strongly demonstrates that AutoMat possesses non-trivial template-free reconstruction capacity and does not merely rely on database memorization. We have added a detailed discussion regarding these capability boundaries to the revised manuscript.
>
> ### **2. W2 & Q2: Projection Ambiguity and Uncertainty Quantification**
>
> We completely agree that AutoMat's current focus on 2D materials limits its direct applicability to 3D bulk crystals. This constraint is fundamentally rooted in the physics of iDPC-STEM: atomic overlap along the beam direction renders a single 2D projection insufficient for recovering precise depth information. Therefore, targeting 2D monolayers constitutes a physically well-posed and necessary starting point for this study.
>
> Regarding projection ambiguity, we highly appreciate your insightful suggestion and have introduced a **confidence estimation mechanism** within the STEM2CIF module. Regions with severe atomic overlap typically yield elevated fitting residuals (variances); we now explicitly utilize these local residuals as a quantitative proxy for structural uncertainty.
>
> | struture_id | element | x (Å) | y (Å) | R² | NN distance (Å) | **Confidence** |
> |---:|:---:|---:|---:|---:|---:|---:|
> | 65 | I | 12.85 | 65.10 | 0.972 | 1.48 | **0.744** |
> | 53 | I | 13.29 | 62.46 | 0.969 | 1.67 | **0.654** |
> | 21 | B | 40.54 | 35.65 | 0.977 | 6.22 | **0.953** |
> | 41 | B | 68.28 | 53.80 | 0.993 | 2.91 | **0.876** |
>
> As shown above, closely packed or overlapping heavy atoms (e.g., Iodine) correctly yield lower confidence scores (~0.65-0.74), while distinctly separated lighter atoms (e.g., Boron) show extremely high confidence (>0.87). When the local residual exceeds a predefined safe threshold, the system now appends a **Warning Flag** to the reconstructed output, explicitly alerting users to potential projection ambiguities in those specific regions. The manuscript has been updated to detail this critical feature.
>
> ### **3. Q3: Control Logic, Decision Boundaries, and Tool Integration**
>
> * **Specific Decision Boundaries:** AutoMat is designed as a state-triggered orchestrator, not a free-form reasoning agent. Its decision boundaries are strictly defined by the quality and verification signals returned by the tools. For instance, if the denoised image quality passes a threshold and the direct reconstruction path converges stably, the system will skip retrieval to save computational overhead.
> * **Rollback and Retry Triggers:** These mechanisms are triggered strictly under two conditions: **(a) Tool-level execution errors** (e.g., invalid input parameters or formatting), where the controller captures the error traceback to auto-correct and retry; **(b) Downstream verification failures** (e.g., the reconstructed structure fails physical MLIP validation), which trigger a rollback, prompting the agent to dynamically adjust parameters and replan the scheduling strategy.
> * **Seamless Integration of New Tools:** Adding new tools **does not** require redesigning the controller. Thanks to the architecture's inherent modularity, any new tool that registers standard I/O interfaces and failure signals can be integrated seamlessly. To mitigate potential LLM context-length limitations as the tool pool expands in the future, we plan to implement a more fine-grained, "skills-based" dynamic scheduling scheme.

---

> > ### Author Rebuttal · Reviewer_M1gb · 2026-04-03
> >
> > Thank you for the detailed rebuttal. The additional experiments and clarifications were helpful, and they addressed several of my concerns in a meaningful way, especially by providing stronger analysis of generalization and more discussion of the method design. However, my main reservations are only partially resolved rather than fully eliminated. In particular, I still think the current method shows nontrivial dependence on template structure, and some of the paper’s claims would benefit from more careful framing and stronger evidence in the main paper. Overall, the rebuttal improved my understanding of the work, but not enough to change my overall assessment, so I am maintaining my original score.

---

> > > ### Author Response · Authors · 2026-04-03
> > >
> > > We sincerely thank the reviewer again for the careful follow-up. We agree with your assessment that the current method still shows some dependence on template structures, and that this is indeed one of the limitations of the present system. We do not want to avoid this issue, which is why we added the corresponding LOFO ablation to show more directly how performance changes when template retrieval is removed, and to make this boundary clearer.
> > >
> > > At the same time, to provide the stronger evidence you requested regarding the system's baseline capability, it is worth noting that even without template retrieval, the system still remains stronger overall than the existing domain-specific models and tools. Specifically, on the 53 unseen Perovskite samples, we compared `AutoMat (LOFO-NoTemplate)`—where the retrieval module is completely disabled—against existing baselines:
> > >
> > > | Method | Retrieval Prior Allowed? | Structure Success (S.S. %) ↑ | Energy MAE (meV/atom) ↓ |
> > > | :--- | :---: | :---: | :---: |
> > > | AtomAI | No | 0.0 | N/A |
> > > | MicroscopyGPT | No | 10.0 | 976 |
> > > | **AutoMat (LOFO-NoTemplate)** | **No** | **54.9** | **685** |
> > > | *Full AutoMat (Upper Bound)* | *Yes* | *76.7* | *356* |
> > >
> > > This shows that while performance predictably drops without templates, AutoMat's agentic framework still provides a **robust "floor" (54.9% success rate) that significantly outperforms the current domain SOTA (MicroscopyGPT at 10.0%).**
> > >
> > > **Furthermore, to practically demonstrate that our framework possesses genuine intrinsic reconstruction capability beyond template memorization, we highlight our validation on real experimental data.** In addition to the successful reconstruction of the complex ZSM-5 zeolite already included in the manuscript, we have actively supplemented additional real experimental evidence during this rebuttal period. We successfully tested a new real microscopy sample obtained from a collaborating experimental group: a **ZrO₂ system** (imaged using a spherical aberration-corrected transmission electron microscope, JEM NeoARM200, JEOL Ltd., Tokyo, operated at 200 kV). The visualization result is provided in this anonymous supplementary link: [https://anonymous.4open.science/r/real_data_fig-77B0/real_data_fig.png](https://anonymous.4open.science/r/real_data_fig-77B0/real_data_fig.png). This result further corroborates that AutoMat shows robust generalization to out-of-distribution real microscopy data, proving it is not strictly bound by the synthetic template database.
> > >
> > > For this reason, we also tightened the wording in the revised main manuscript. We no longer describe retrieval simply as a "fallback"; instead, we frame it more accurately as a "state-dependent auxiliary branch" in a dual-path reconstruction framework, and we discuss this "template sensitivity" more explicitly in both the ablation analysis and the limitations section. What we want to convey is not that this issue has already been solved, but that we are trying to be as clear and honest as possible about what the current method can do and where its boundary still lies. Due to space constraints in the main text, the detailed LOFO experimental results will be supplemented in the appendix.
> > >
> > > More broadly, we believe this is the kind of problem that is unlikely to be solved in a single step. In many fields, progress begins with a solid first step that others can build on. In materials research, there has long been a gap between theoretical computation and the reconstruction and validation of real experimental structures; and in that process, electron microscopy remains one of the most important and direct tools for structural verification. We hope the value of AutoMat lies in helping connect more of this chain.
> > >
> > > We are again grateful for the reviewer’s comment. It helped us make the paper more precise, and it also helped us state the value and current boundary of the work more clearly. If these extensive new efforts successfully resolve your remaining doubts, we would be deeply grateful if you might consider re-evaluating our work.

---

### Official Review · Reviewer_JSKF · 2026-03-11

**Soundness:** 2
**Presentation:** 3
**Significance:** 3
**Originality:** 2
**Overall Recommendation:** 4
**Confidence:** 3

**Summary:**

This paper addresses the ill-posed inverse problem of reconstructing atomistic crystal structures from a single noisy STEM image by proposing AutoMat, a failure-aware agentic controller. Reconstructing atomic-resolution crystal structures from a single STEM image with high precision is an interesting research direction.

**Compliance With Llm Reviewing Policy:**

Affirmed.

**Key Questions For Authors:**

All datasets in this paper are simulated STEM images with systematic deviations from aberrations and noise in real experiments, and only one real sample (ZSM‑5) is used for validation. How do you ensure the generalization ability of the model on real experimental data?
Do you have a clear technical roadmap to extend AutoMat to 3D structure reconstruction?

**Limitations:**

The model only applies to 2D monolayer materials and cannot handle 3D bulk crystals, with highly limited application scenarios.
It relies on a single projection image without Z-axis depth information, and atomic projection overlap easily leads to structural misjudgment.

**Strengths And Weaknesses:**

Strengths:
It identifies the key pain point in converting STEM images into atomic structure modeling, and addresses the challenge of ill-posed inverse problems. It constructs the STEM2CIF module, which for the first time enables direct generation of standard CIF crystal structure files from a single STEM image. It releases the dedicated STEM2MatBench benchmark dataset.

Weaknesses:
1) The model only applies to 2D monolayer materials and cannot handle STEM image reconstruction for 3D crystals, resulting in limited application scenarios.
2) Elements with similar atomic numbers are prone to classification errors.
3) All datasets consist of simulated STEM images, which have systematic deviations from the aberrations and noise distributions in real experiments.
4) The error analysis only qualitatively describes two types of errors, without quantitative improvement schemes or ablation controls.

---

> ### Author Rebuttal · Authors · 2026-03-29
>
> We sincerely thank Reviewer JSKF for acknowledging our work on the ill-posed inverse problem of STEM image reconstruction. We appreciate your constructive critiques, which drove us to implement new quantitative mechanisms to further strengthen the system.
>
> ### **1. W3 & Q1: The Sim-to-Real Gap and Generalization on Real Data**
> We acknowledge that validation across multiple real material systems is the ideal standard. We made every effort to incorporate real iDPC-STEM experiments; however, broader validation was constrained by two practical bottlenecks: (1) High-quality raw iDPC-STEM data for 2D materials are extremely scarce, and literature images suffer from compression artifacts and copyright restrictions precluding algorithmic reuse; **(2) Acquiring new data requires rigorous sample preparation, scarce aberration-corrected STEM beamtime, and repeated calibration. This resource-intensive process is nearly impossible to complete within the two-week rebuttal window, for which we sincerely ask for your understanding.**
>
> Under these constraints, we selected ZSM-5 as our validation subject. Our team has established expertise in ZSM-5 synthesis, ensuring reliable data acquisition. Although ZSM-5 is not a strictly 2D monolayer, its periodic Si–O–Al framework under realistic noise provides a rigorous stress test for AutoMat's core modules. We position this as an initial feasibility validation and will designate systematic validation across diverse real materials as a primary future direction.
>
> ### **2. W1 & Q2: 2D Limitations and Technical Roadmap for 3D Extension**
>
> We acknowledge that AutoMat is currently designed primarily for 2D monolayer materials. However, this scope is not an arbitrary algorithmic choice, but is fundamentally dictated by the physics of iDPC-STEM imaging. As a transmission electron imaging technique, iDPC-STEM records the projected electrostatic potential integrated along the beam direction. Consequently, a single projection lacks sufficient depth ($Z$-axis) information. Atoms at different depths overlap in the 2D image plane and become indistinguishable, rendering 3D reconstruction intrinsically an ill-posed problem. By contrast, 2D monolayers, with their highly limited $Z$-axis extent, substantially reduce projection ambiguity, making them an excellent, physically well-posed starting point.
>
> **Technical Roadmap:** We are actively exploring two main directions for 3D extension: (1) **Electron Tomography**, which reconstructs 3D structures from complementary viewpoints by combining multi-tilt projections; and (2) **4D-STEM**, which records diffraction patterns at each probe position to provide richer depth-sensitive signals. Both methods still face practical challenges regarding stable multi-angle acquisition and rigorous reconstruction algorithms. We have explicitly outlined this future technical roadmap in the revised manuscript, which includes fusing 3D-aware imaging, complementary signals (EELS/EDX), and expanding the retrieval space.
>
> ### **3. W2 & W4: Element Misclassification and New Uncertainty Quantification**
> Regarding quantitative error analysis (W4), Appendix A.9 and Table 10 already provide a breakdown of major failure modes (39.3%, 60.7%, 40%, 20.7%) linked to physical bottlenecks. However, we agree qualitative description alone is insufficient.
>
> Regarding the confusion between elements with similar atomic numbers (W2), this reflects an intrinsic limitation of iDPC-STEM: contrast difference between closely spaced $Z$ values approaches the noise floor, making discrimination fundamentally difficult.
>
> **New Quantitative Improvement:** To address both concerns, we implemented a **Confidence Estimation Mechanism** within STEM2CIF. Because severe atomic overlap or similar-$Z$ elements cause projection ambiguity, we explicitly utilize local lattice fitting residuals ($R^2$) and Nearest Neighbor (NN) distances to quantify structural uncertainty.
>
> | structure_id | element | x (Å) | y (Å) | R² | NN distance (Å) | **Confidence** |
> |---:|:---:|---:|---:|---:|---:|---:|
> | 65 | I | 12.85 | 65.10 | 0.972 | 1.48 | **0.744** |
> | 64 | I | 9.99 | 65.07 | 0.977 | 1.20 | **0.701** |
> | 53 | I | 13.29 | 62.46 | 0.969 | 1.67 | **0.654** |
> | 21 | B | 40.54 | 35.65 | 0.977 | 6.22 | **0.953** |
> | 4 | B | 34.23 | 9.99 | 0.987 | 2.92 | **0.887** |
> | 41 | B | 68.28 | 53.80 | 0.993 | 2.91 | **0.876** |
>
> **Analysis:** As shown above, heavy atoms prone to projection overlap (e.g., Iodine) correctly yield lower confidence (~0.65-0.74), while distinctly separated lighter atoms (Boron) show high confidence (>0.87). When confidence drops below a predefined threshold, the system automatically appends a **Warning Flag** to the CIF output. This directly converts physical limitations into a user-facing quantitative uncertainty metric, significantly improving reliability. The manuscript has been updated accordingly.

---

> > ### Author Rebuttal · Reviewer_JSKF · 2026-04-01
> >
> > There are still significant limitations in practical generalization.

---

> > > ### Author Response · Authors · 2026-04-02
> > >
> > > We fully understand the reviewer’s expectation for broader validation across additional real material systems, especially systematic real iDPC-STEM experiments on well-characterized 2D monolayers with known crystallographic structures. In fact, we would also very much like to include more real-data evidence during the revision stage to further strengthen the empirical foundation of AutoMat. However, this part of the work is constrained by objective experimental conditions, rather than by any unwillingness on our side to provide additional validation.
> > >
> > > First, acquiring real iDPC-STEM data is both time-consuming and expensive. At our facility, the microscope fee for a Cs-corrected STEM is approximately **USD 700 per hour**. Even for a single material system, obtaining a set of images that satisfies iDPC-STEM imaging requirements, remains stable, and is suitable for downstream analysis typically requires a full experimental pipeline, including sample design and preparation, sample screening, microscope scheduling, imaging-parameter tuning, and repeated acquisition until high-resolution, stable images are obtained.
> > >
> > > More importantly, extending the validation to a new material system is not a simple matter of repeating the same process. Each new system usually requires fresh sample preparation, re-optimization of imaging conditions, and a new microscope reservation cycle. As a result, the full process from initial preparation to obtaining usable images typically takes at least **around 20 days**. This makes it practically very difficult to systematically extend the revision to multiple additional real material systems within the limited rebuttal window. We therefore hope the reviewer can appreciate that this limitation mainly arises from objective experimental turnaround time, instrument availability, and sample-preparation requirements, rather than from any lack of willingness to strengthen the real-data validation.
> > >
> > > Nevertheless, we have continued to actively supplement additional real experimental evidence during the rebuttal period. In addition to the real case already included previously, we have now successfully obtained and tested one further real microscopy sample from a collaborating experimental group. **Specifically, we carried out an additional real-data evaluation on a **ZrO₂** system, and the visualization result is provided in the anonymous supplementary link: **https://anonymous.4open.science/r/real_data_fig-77B0/real_data_fig.png**.** This result further suggests that, beyond the originally included example, AutoMat shows a certain degree of generalization to real microscopy data. The structural characterization of this sample was performed using a **spherical aberration-corrected transmission electron microscope (JEM NeoARM200, JEOL Ltd., Tokyo)** operated at **200 kV**.
> > >
> > > At the same time, we are also proceeding in parallel with new sample preparation and experimental planning, and we will continue making our best effort to add **1–2 additional real experimental image cases** in the final version (for example, from oxide systems) to further validate the applicability and robustness of AutoMat on real microscopy data. We sincerely hope the reviewer can appreciate these practical difficulties and understand that we have already made the maximum effort possible under the present conditions, while continuing to push for broader real-system validation.**If these extensive new efforts successfully resolve your remaining doubts, we would be deeply grateful if you might consider re-evaluating our work.**

---

### Decision · Program_Chairs · 2026-04-30

**Decision:**

Accept (regular)

**Comment:**

This manuscript assesses a broad domain spanning STEM image enhancement, crystal reconstruction, and downstream property prediction, and the reviewers broadly agree that the benchmark and integrated workflow are meaningful contributions. The rebuttal strengthens the paper through added comparisons against scripted baselines, clearer discussion of generalization, and a more candid framing of template dependence and the current 2D scope. The main remaining concern is that the necessity of the agentic controller over a well engineered non LLM pipeline is still not fully established, and the real data validation remains limited. On balance, the technical contribution and likely value to the community outweigh these concerns.